# New enantiornithine diversity in the Hell Creek Formation and the functional morphology of the avisaurid tarsometatarsus

**Alexander D. Clark**[1,2]*, **Jessie Atterholt**[3], **John B. Scannella**[4], **Nathan Carroll**[5], **Jingmai K. O'Connor**[2]

**1** Committee on Evolutionary Biology, University of Chicago, Chicago, IL, United States of America, **2** Negaunee Integrative Resource Center, Field Museum of Natural History, Chicago, IL, United States of America, **3** College of Osteopathic Medicine of the Pacific, Western University of Health Sciences, Pomona, CA, United States of America, **4** Museum of the Rockies and Department of Earth Sciences, Montana State University, Bozeman, Montana, United States of America, **5** Carter County Museum, Carter County, MT, United States of America

* adclark@uchicago.edu

**Data Availability Statement:** All data used for analyses is present within either in-text tables or supplementary tables. Additionally, coding has

## Abstract

Enantiornithines were the most diverse group of birds during the Cretaceous, comprising over half of all known species from this period. The fossil record and subsequently our knowledge of this clade is heavily skewed by the wealth of material from Lower Cretaceous deposits in China. In contrast, specimens from Upper Cretaceous deposits are rare and typically fragmentary, yet critical for understanding the extinction of this clade across the K-Pg boundary. The most complete North American Late Cretaceous enantiornithine is *Mirarce eatoni*, a member of the diverse clade Avisauridae. Except for *Mirarce*, avisaurids are known only from isolated hindlimb elements from North and South America. Here we describe three new enantiornithines from the Maastrichtian Hell Creek Formation, two of which represent new avisaurid taxa. These materials represent a substantial increase in the known diversity of Enantiornithes in the latest Cretaceous. Re-examination of material referred to Avisauridae through phylogenetic analysis provides strong support for a more exclusive Avisauridae consisting of six taxa. Exploration of the functional morphology of the avisaurid tarsometatarsus indicates potential strong constriction and raptorial attributes. The lower aspect ratio of the tarsometatarsus facilitates a more biomechanically efficient lever system which in extant birds of prey equates to lifting proportionally heavier prey items. In addition, the proportional size and distal position of the *m. tibialis cranialis* tubercle of the tarsometatarsus is similar to the morphology seen in extant birds of prey. Together with the deeply-grooved metatarsal trochlea facilitating robust and likely powerful pedal digits, morphologies of the hindlimb suggest avisaurids as Late Cretaceous birds of prey.

## Introduction

Our understanding of Mesozoic avian diversity and evolution is limited by available material, increasing with each new described taxon. Several Lower Cretaceous Lagerstätte have yielded

been included that produced our discussed quantitative results.

**Funding:** The author(s) received no specific funding for this work.

**Competing interests:** The authors have declared that no competing interests exist.

abundant fossils which provide insights to the diversity, ecology, and taxonomy of Mesozoic birds relatively early in their evolution [1]. However, information from Upper Cretaceous deposits is far more limited, with the only possible productive bonebed being a newly emerging Brazilian site dubbed "William's Quarry" [2, 3]. Late Cretaceous fossil birds are critical for understanding one of the most important aspects of avian evolution: the selectivity of the end Cretaceous Mass Extinction in which only neornithines survived.

The most speciose clade of Mesozoic birds, constituting over half of all described species, are the enantiornithines [4–6]. Members of this clade exhibit morphological diversity and are found on every continent except for Antarctica and continental Africa [4–7]. The vast majority of known specimens and described taxa come from Hauterivian-Aptian Lower Cretaceous deposits in Spain and China. As a result, our knowledge of this clade is heavily biased to a narrow time interval of approximately 10 million years [1, 8–11].

Enantiornithine fossils from the Late Cretaceous are far more limited but critical for understanding the extinction of this once successful lineage. The only widely recognized clade from Upper Cretaceous deposits is the Avisauridae, known exclusively from North and South American deposits 86–66 Ma. Six genera have been named in the literature: *Avisaurus* [12], *Soroavisaurus* [13], *Neuquenornis* [14], *Intiornis* [15], *Mirarce* [16], and *Gettyia* (formerly *A. gloria)* [16, 17]. Members of this group are mostly described from fragmentary remains, with the tarsometatarsus being the only commonly recovered element and thus critical for recognizing members of this clade.

Previously, three synapomorphies were hypothesized to unite Avisauridae: a strongly convex dorsal margin of metatarsal III; the plantar projection of the medial condyle of metatarsal III; and a J-shaped laterally compressed metatarsal I [14]. One cladistic analysis suggested this clade also included *Enantiophoenix*, *Halimornis* and *Concornis* [18]. However, support for this hypothesis is weak. The tarsometatarsus is unknown in *Halimornis*, and thus referral to the Avisauridae is ambiguous. Similarly, the tarsometatarsus of *Enantiophoenix* is poorly preserved, making the assignment to the Avisauridae based on the convexity of the dorsal surface equivocal. Convexity of the dorsal margin of metatarsal III appears to be more widespread than previously thought, being present in non-avisaurid enantiornithines (e.g., *Imparavis*, *Feitianius*) [19, 20]. Additionally, as in *Enantiophoenix*, the lack of convexity in other enantiornithines may be taphonomic.

Assignment to the Avisauridae for *Concornis* was also based on the convexity of the dorsal margin of metatarsal III, and stoutness of the first phalanx of pedal digit I [18]. In this study, *Halimornis* and *Concornis* were regarded as distinct lineages within the Avisauridae, although with little morphological support [18].

The taxon from which this family derives its name, *Avisaurus archibaldi*, is a large enantiornithine, with a tarsometatarsal length (~ 75 mm) comparable to large extant larids (e.g., *Larus marinus*, *L. glaucescens*) and accipitrids (e.g., *Buteo jamaicensis*, *Accipiter gentilis*) [12, 21, 22]. This is in stark contrast to nearly all Early Cretaceous enantiornithines, whose average tarsometatarsal length varies between extant emberizids and turdids, tending not to exceed 25 mm in length [4] (S1 Table). Avisaurids not only add to the morphological diversity of enantiornithines, but also contribute considerably to our understanding of the apparent trend of increasing body size diversity in the Late Cretaceous [4, 12, 13, 16].

Here we describe three new enantiornithine specimens, all of which come from the Hell Creek Formation (Latest Cretaceous 68–66 Ma) [23]. One of these new specimens is considered a new species within the genus *Avisaurus* and another is assigned to *Avisaurus* sp.. Previously, *A. archibaldi* was the only avisaurid and enantiornithine described from these deposits [12]. Like previously described North American avisaurids except for *Mirarce*, the new specimens are all represented by isolated tarsometatarsi. Although the material is limited, they

possess morphological differences from previously known Late Cretaceous enantiornithines. Here, we revise morphological and phylogenetic diagnoses for the Avisauridae and discuss the significance of these new specimens with regards to their phylogenetic position, implications for size diversity among enantiornithines, and the functional morphology of the avisaurid tarsometatarsus.

INSTITUTIONAL ABBREVIATIONS–DDM, Dinosaur Discovery Museum Kenosha, Wisconsin, United States; MOR, Museum of the Rockies Bozeman, Montana, United States; CCM, Carter County Museum Ekalaka, Montana, United States.

## Methods

All necessary permits were obtained for the described study, which complied with all relevant regulations. Specimens were photographed under normal lighting conditions with a Canon 70D dslr camera with a Canon EF 100mm f/2.8 macro lens. Photos of each specimen were assessed in Adobe Photoshop version 24.3.0 and if needed, had aspects of contrast, and color balance edited. Anatomical nomenclature primarily follows Baumel and Witmer [24] using the English equivalents of the Latin terminology. All measurements used for description and comparisons were recorded in-person using digital calipers (OriginCal1P54). Additional measurements were recorded with ImageJ [25]. Aspect ratios of tarsometatarsi were calculated by dividing the mediolateral width at the proximodistal midpoint by the total proximodistal length at the mediolateral midpoint of the proximal-most margin to the distal-most margin of the proximodistally longest metatarsal.

Size estimates for three avisaurids (*Mirarce*, DDM 1577.730, *A. archibaldi*) were calculated to better compare body sizes to other enantiornithines and extant birds. Even though the tarsometatarsus scales with size in extant birds (S1 Fig), Field et al. [26] found the greatest predictive power (highest $R^2$ scores) for determining the mass of birds was the maximum diameter of the glenoid facet of the coracoid (maximum diameter of the coracoid's humeral articulation facet–HAF) and the length of the humerus, both which can be measured in the avisaurid *Mirarce*. To estimate the mass of the two larger avisaurids known only from tarsometatarsi (DDM 1577.730 and *A. archibaldi*) the diameter of the coracoid glenoid and the length of the humerus were approximated by scaling them from *Mirarce* by using a bone which all three taxa preserve: the tarsometatarsus. The tarsometatarsus of DDM 1577.730 and *A. archibaldi* measure 140% and 155% that of *Mirarce* respectively. Using these percentages, we scaled the diameter of the coracoid glenoid and the length of the humerus in *Mirarce* to estimated lengths for DDM 1577.730 and *A. archibaldi* to allow for mass predictions. For all three avisaurids, we averaged the masses of the nearest ten extant species according to data collected by Field et al. [26] using both the diameter of the coracoid glenoid and the length of the humerus.

### Nomenclatural acts

The electronic edition of this article conforms to the requirements of the amended International Code of Zoological Nomenclature, and hence the new names contained herein are available under that Code from the electronic edition of this article. This published work and the nomenclatural acts it contains have been registered in ZooBank, the online registration system for the ICZN. The ZooBank LSIDs (Life Science Identifiers) can be resolved and the associated information viewed through any standard web browser by appending the LSID to the prefix ""http://zoobank.org/"".

The LSID for this publication is: urn:lsid:zoobank.org:pub:53881EA9-CD01-4EAD-9339-ED912FE5C952

The electronic edition of this work was published in a journal with an ISSN, and has been archived and is available from the following digital repositories: PubMed Central, LOCKSS.

## Systematic paleontology

Aves Linnaeus, 1758
Pygostylia Chiappe, 2002
Ornithothoraces Chiappe, 1995
Enantiornithes Walker, 1981
Avisauridae Brett-Surman and Paul, 1985

## Diagnosis

Enantiornithines characterized by the combination of a robust tarsometatarsus (mediolateral width at the midpoint at least 20% that of the total proximodistal length); a mediolaterally broad trochlea of metatarsal II relative to III; the presence of a hypertrophied tubercle for the attachment of the *m. tibialis cranialis* on metatarsal II; the *m. tibialis cranialis* tubercle located at least 20% down the length of the tarsometatarsus from the proximal margin; and a well-developed plantar projection of the medial margin of the trochlea of metatarsal III (modified from Atterholt et al., [18]).

## Included genera

*Avisaurus*–UCMP 117600, Hell Creek Formation, Late Cretaceous, Maatstrichtian, Garfield County, Montana [12]; *Soroavisaurus*–PVL 4690, Lecho Formation, Late Cretaceous, Maastrichtian, Salta Province, Argentina [13]; *Mirarce*–UCMP 139500, Kaiparowits Formation, Late Cretaceous, Campanian, Grand Staircase-Escalante National Monument, Utah [16]; *Gettyia*–MOR 553E/1.19.91.64, Two Medicine Formation Late Cretaceous, Campanian, Glacier County, Montana [16].

## *Avisaurus* Brett-Surman and Paul, 1985

**Diagnosis.** Avisaurid enantiornithines characterized by the following combination of characters: distal extent of metatarsal II approximately equal to that of metatarsal IV; the proximoplantar surface excavated by a pair of small fossae located where the medial and lateral margins of metatarsal III meet the plantar labrum, hereafter referred to as the proximoplantar fossae (*nomen novum*) (modified from Chiappe, [13]) (S2 Fig).

**Included species.** *Avisaurus archibaldi*–UCMP 117600, Late Cretaceous, Maatstrichtian, Garfield County, Montana [12]; *Avisaurus darwini*–DDM 1577.730, Hell Creek Formation, Late Cretaceous, Maastrichtian, Carter County, Montana (this publication).

*Avisaurus darwini* **sp. nov. (Fig 1 and Table 1).**

**Holotype.** DDM 1577.730 is a complete right tarsometatarsus collected in July of 2022

**Etymology.** The specific name "*darwini*" is in honor of Charles Darwin, whose momentous research and publications helped define the field of evolutionary biology. *Avisaurus darwini*, Darwin's bird lizard.

Type horizon and locality: Hell Creek Formation, Latest Cretaceous, Maastrichtian (approximately 66 Ma). DDM 1577.730 was collected in Carter County, Montana at a locality under the auspices of the Bureau of Land management. The specimen was collected from the Madison Shark Spine locality (DDM-18-234,1N 55E 11) within a grey sand multitaxic microvertebrate site.

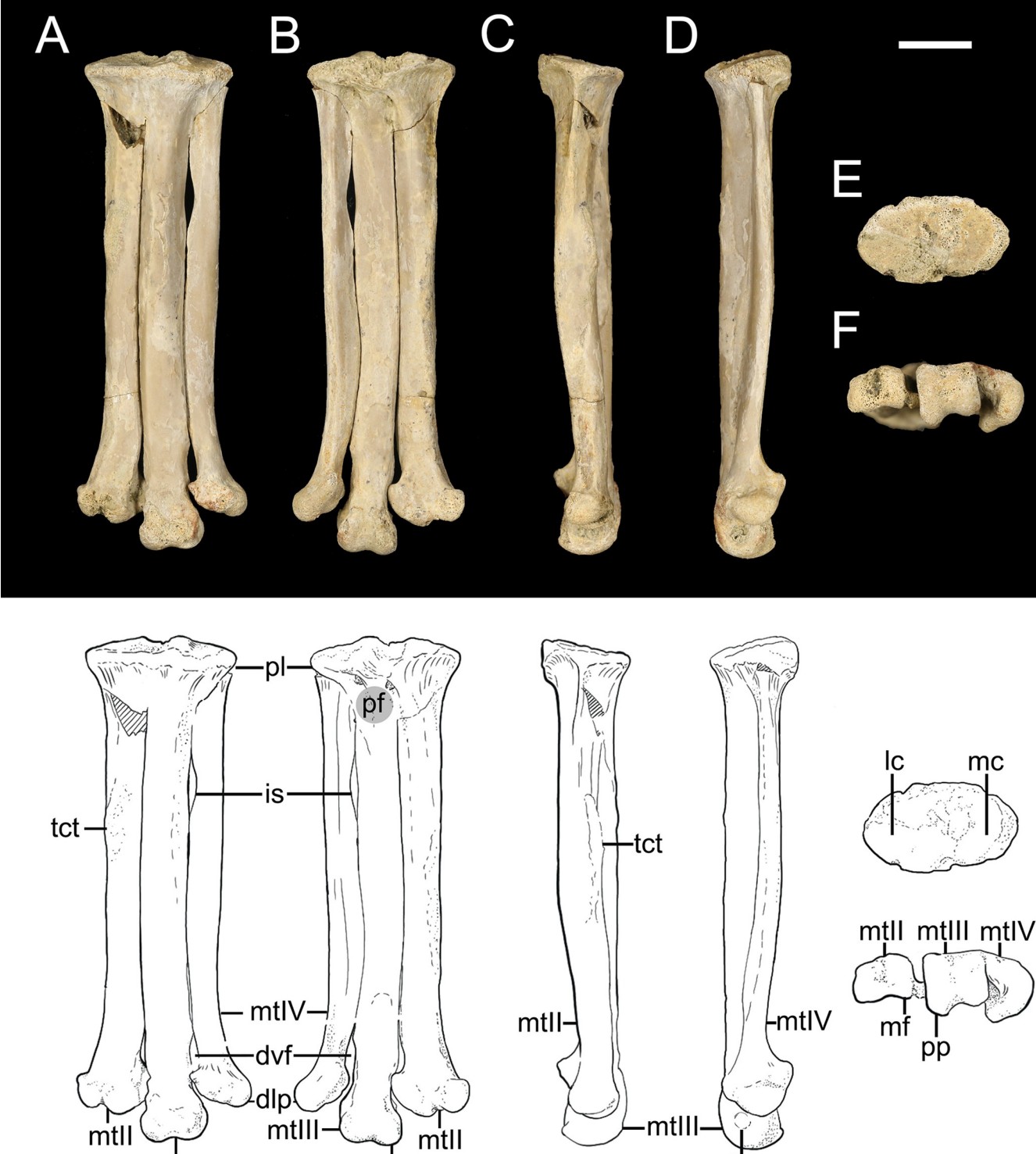

**Fig 1.** The left tarsometatarsus of *Avisaurus darwini* (DDM 1577.730) in A) dorsal, B) plantar, C) right lateral, D) left lateral, E) proximal, and F) distal aspects. Abbreviations: dlp, distal lateral projection; dmp, distal medial projection; dvf, distal vascular foramen; fo, fossa (collateral ligament fossa); is, intermetatarsalian space; lc, lateral cotyle; lf, laterally-projecting flange; mc, medial cotyle; mtII, metatarsal II; mtIII, metatarsal III; mtIV, metatarsal IV; pf, proximoplantar fossae; pl, proximal labrum; pp, plantar projection; tct, *m. tibialis cranialis* tubercle. Scale bar 10mm. Line illustrations done by Samantha Clark.

**Table 1. Measurements (in mm) of each newly described specimen.** For MOR 3070, asterisks represent approximations. Total length of metatarsal II was estimated by scaling MOR 3070 to *A. darwini*, which is nearly the same size.

|  | *Avisaurus darwini* | *MOR 3070* | *Magnusavis ekalakaensis* |
|---|---|---|---|
| **Proximal Surface** |  |  |  |
| Width | 20.24 | 18.56 | - |
| Height | 11.65 | 11.15 | - |
| **Cranial Aspect** |  |  |  |
| MT II Length | 62.93 | *58.68 | - |
| MT II Width (midpoint) | 4.91 | - | 3.12 |
| MTIII Length | 65.38 | - | - |
| MT III Width (midpoint) | 6 | - | 4.8 |
| MT IV Length | 60.4 | - | - |
| MT IV Width (midpoint) | 4.98 | - | - |
| **Distal Aspect** |  |  |  |
| MT II Trochlea Width | 8.07 | - | 7.38 |
| MT II Trochlea Height | 5.22 | - | 5.78 |
| MT III Trochlea Width | 8.18 | - | 8.27 |
| MT III Trochlea Height | 6.62 | - | 5.87 |
| MT IV Trochlea Width | 9.48 | - | - |
| MT IV Trochlea Height | 4.64 | - | - |
| **Aspect ratio of Tmt** | 1:4 | *1:4 | - |

Diagnosis–A large accipitrid-sized bird identified as an avisaurid based on the strong plantar projection of the medial condyle of the trochlea of metatarsal III; a robust tarsometatarsus (mediolateral width at the midpoint greater than 20% that of the total proximodistal length); a mediolaterally broad trochlea of metatarsal II relative to III; the presence of a hypertrophied *m. tibialis cranialis* tubercle on metatarsal II located ~ 30% down the tarsometatarsus. DDM 1577.730 is referable to the genus *Avisaurus* based on the following features: the distal extent of metatarsal II approximately equal to that of metatarsal IV; the presence of proximoplantar fossae; a metatarsal II exhibiting a laterally-projecting flange of bone from the plantar margin. DDM 1577.730 can be differentiated from other avisaurids based on the following unique combination of features: a proximally located intermetatarsal space between metatarsals III and IV which is proximodistally proportionally shorter than that of *Soroavisaurus*; metatarsal II's laterally-projecting flange of bone closes the intertrochlear incisure, contacting the distomedial margin of the metatarsal III trochlea (this feature is more pronounced than that in the lateroplantarly-projecting flange in *A. archibaldi*); both medial and lateral margins of the trochlea of metatarsal III terminate distally at the same level, disparate from *Soroavisaurus* (lateral margin extending distally) and *A. archibaldi* (medial margin extending distally).

### *Avisaurus* sp. (Fig 2 and Table 1)

**Material.** MOR 3070 is a partial right tarsometatarsus preserving most of metatarsal II (distal portion disarticulated), < 50% of metatarsal III, and the proximal portion of metatarsal IV. Some additional disarticulated fragments are probably referable to the specimen but lack diagnostic morphologies.

**Locality.** Hell Creek Formation, Latest Cretaceous, Maastrichtian (approximately 68–66 Ma). MOR locality no. HC-720 "Daigo's Site". MOR 3070 was discovered by Daigo Yamamura in Makoshika State Park, Glendive, Montana and collected by a field crew led by Frankie Jackson of Montana State University [27].

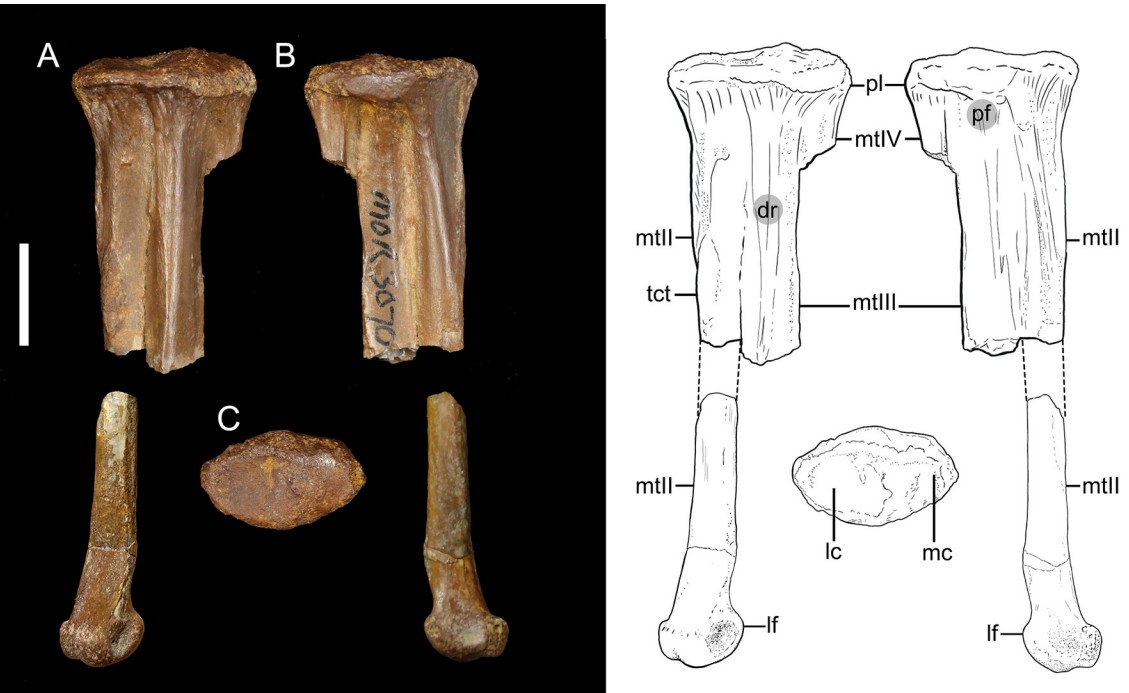

**Fig 2. The left partial tarsometatarsus of *Avisaurus* sp.** MOR3070 in A) dorsal, B) plantar, and C) proximal aspects. The distal portion of metatarsal II is placed in the approximate place if it was articulated. Abbreviations: dr, dorsal ridge: lc, lateral cotyle; lf, laterally-projecting flange; mc, medial cotyle; mtII, metatarsal II; mtIII, metatarsal III; mtIV, metatarsal IV; pf, proximoplantar fossae; pl, proximal labrum; tct, *m. tibialis cranialis* tubercle. Scale bar 10mm. Line illustrations done by Samantha Clark.

Comment–A large accipitrid-sized enantiornithine with plantar-projecting medial and lateral plantar crests; dorsal face of metatarsal II is mediolaterally wider than metatarsal III proximally; the lateral cotyle elevated slightly proximally relative to the medial cotyle. MOR 3070 is referable to the genus *Avisaurus* based on the two proximoplantar fossae in plantar aspect, and a metatarsal II trochlea exhibiting a lateroplantarly-projecting flange of bone from the plantar margin. MOR 3070 differs from other closely related enantiornithines (avisaurids) based on the following unique combination of features: lateral to medial tapering of the proximal labrum's dorsal margin; a dorsally-projecting *m. tibialis cranialis* tubercle rather than dorsomedially projection as in *A. archibaldi* and *A. darwini*; metatarsal II mediolaterally greater than that of III in dorsal aspect along its proximal preserved portion; a well-developed dorsally-projecting ridge on the dorsal face of metatarsal III beginning prominently below the labrum shelf and continuing distally down the dorsal surface of the remaining portions of metatarsal III (forming a triangular cross section in plantar aspect); plantar margin of the trochlea of metatarsal II with a lateroplantarly-projecting flange, similar to *A. archibaldi* but disparate from the lateral-projecting morphology in *A. darwini*.

Aves Linnaeus, 1758
Pygostylia Chiappe, 2002
Ornithothoraces Chiappe, 1995
Enantiornithes Walker, 1981

## *Magnusavis ekalakaenis* gen. et sp. nov. (Fig 3 and Table 1)

**Holotype.** CCM V2019.5.1 is a partial right tarsometatarsus, missing the proximal portions of metatarsals II and II, and all of metatarsal IV. A single phalanx is preserved, though digit assignment is uncertain.

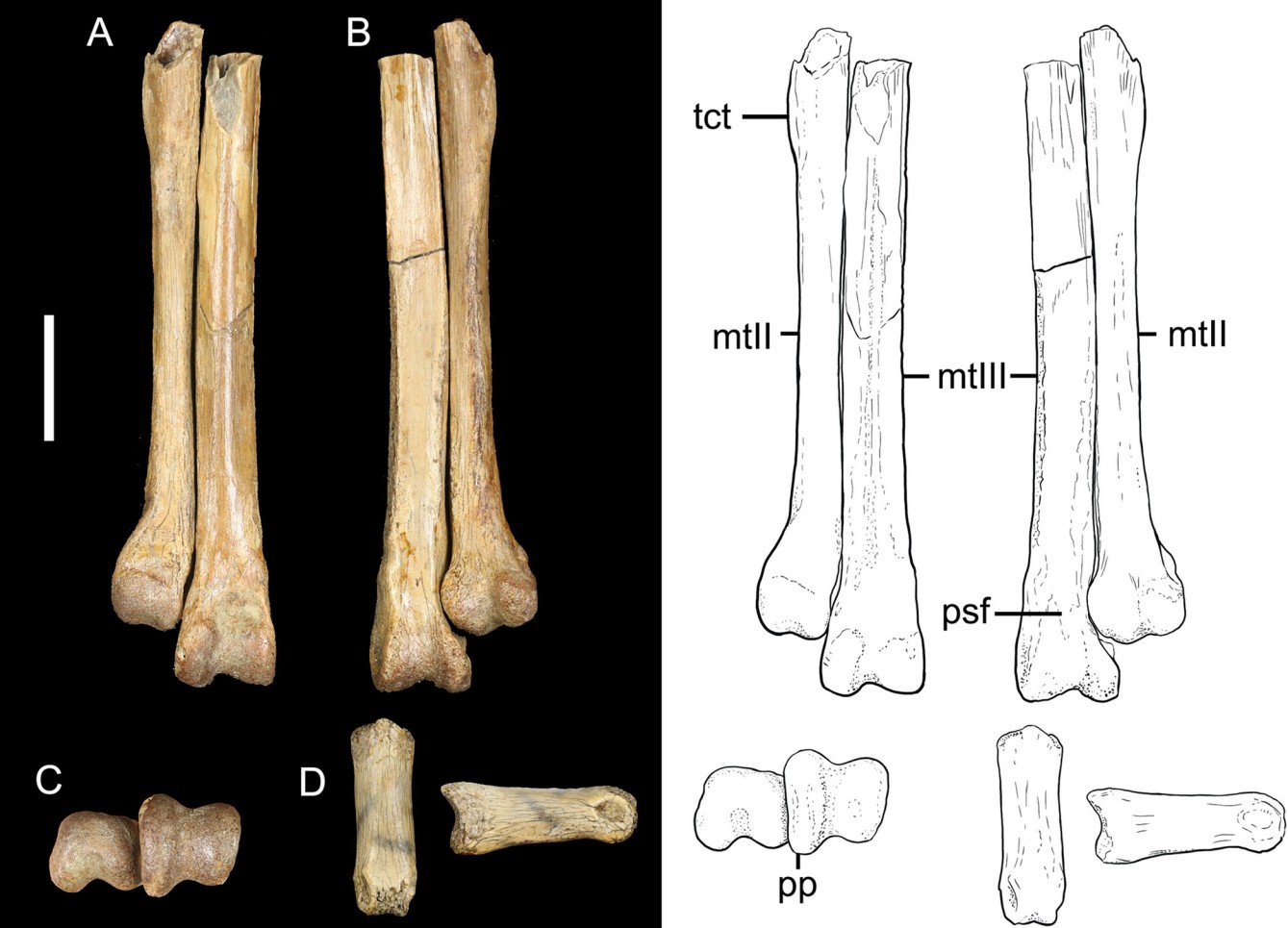

**Fig 3.** The partial left tarsometatarsus of *Magnusavis ekalakaensis* (CCM V2019.5.1) in A) dorsal, B) plantar, and C) distal aspects, D) and the single preserved pedal phalanx in dorsal and right lateral aspects. Abbreviations; mtII, metatarsal II; mtIII, metatarsal III; pp, proximal projection; psf, plantar supratrochanter fossa; tct, *m. tibialis cranialis* tubercle. Scale bar 10mm. Line illustrations done by Samantha Clark.

**Etymology.** In Latin, "*Magnus*" meaning big, and "*avis*" meaning bird, and "*ekalakaensis*" in honor of the town of Ekalaka, Montana, close to where this specimen was discovered. Ekalaka is Lakota for "one who wanders". *Magnusavis ekalakaenis*, Ekalaka's big bird.

**Locality.** Hell Creek Formation, Late Cretaceous, Maastrichtian (approximately 66 Ma). The specimen was collected in Carter County, Montana at a locality under the auspices of the Bureau of Land management (Permit MTM 108846).

Diagnosis–A larid-sized enantiornithine differentiated from other taxa, including the closely-related avisaurids, based on the following unique combination of features: distal expansion of metatarsals II and III (all avisaurids retain nearly equal medial and lateral margins of the metatarsals down their lengths); a medial and lateral-rimmed (i.e., well-excavated) plantar supratrochanter fossa on metatarsal III; and an articular face on metatarsal III (for metatarsal IV) terminating distal to that in *A. archibaldi*.

## Phylogenetic analysis

These three new specimens were added to a modified version of the character matrix by Atterholt et al. [16]. We added three new characters to assess the relative robustness of the

tarsometatarsus [Character 253], presence of the proximoplantar fossae bordering metatarsal III [Character 254], and the presence of the intermetatarsal space between metatarsals III and IV proximal to the midpoint of the tarsometatarsus [Character 255]. The final matrix, consisting of 45 taxa with scores across 255 characters, was analyzed in TNT [28]. For scoring, see Supplementary Information. Neornithes were represented by *Anas* and *Gallus*, and Dromaeosauridae was used as the outgroup. We conducted a heuristic search retaining the single shortest tree out of every 1000 trees followed by a second round of tree-bisection reconnection (TBR). The first round of TBR produced 36 trees with a length of 803 steps; the second round of TBR produced 4207 trees of the same length.

## Results

### Morphological descriptions

DDM 1577.730 –Holotype of *Avisaurus darwini*

urn:lsid:zoobank.org:act:E5235AC4-571C-4606-90D7-C25BCDC3CF2A

The metatarsals are all fused proximally to each other and the distal tarsals. The proximal articular surface is oval in shape with the lateral cotyle slightly elevated and larger relative to the medial cotyle as in *A. archibaldi*. It is greater than the cross-section of metatarsals II-IV such that it forms a well-developed labrum that overhangs the metatarsal shafts in both dorsal and plantar aspects as in *Mirarce* and *A. archibaldi* [12, 16]. Like *A. archibaldi*, this is more pronounced in plantar aspect. In plantar view, just below the labrum, the proximal-most lateral and medial portions of the metatarsal III shaft are each excavated by a small fossa, referred to as the proximoplantar fossae (Fig 1B). These fossae are also present, although less pronounced, in *A. archibaldi* and MOR 3070, but absent in other avisaurids.

In dorsal aspect, the metatarsals do not distally expand mediolaterally, similar to *A. archibaldi*, *Gettyia*, and *Mirarce* [12, 16] (Fig 4). As in *Soroavisaurus*, there is a large intermetatarsal space between metatarsals III and IV (Fig 1A and 1B), originally referred to as a "fenestra" Chiappe [13]. This structure may be analogous to the lateral proximal vascular foramen in extant birds which allows for anastomosis between the dorsal and plantar surfaces [13, 29, 30]. The mediolateral width of the space measures approximately 45% of the midpoint of metatarsal IV. In *Soroavisaurus*, this space is proximodistally longer and begins more proximally, extending from nearly 5% to 36% the length of the tarsometatarsus [13]. In *Intiornis* the space extends from 12% to 30% the length of the tarsometatarsus [15].

Metatarsal III is the longest, as in all avisaurids, and most enantiornithines [4, 16]. The proximodistal length of metatarsals II and IV measures 96% and 92% that of metatarsal III respectively. Metatarsal III also exhibits the greatest mediolateral width measured at the midpoint. Metatarsals II and IV measure 82% and 83% the width of metatarsal III, respectively (Table 1). Like other avisaurids (but also some non-avisaurid enantiornithines), the dorsal surface of metatarsal III is strongly convex forming a weak ridge for most of its length. The shaft of metatarsal II is similarly convex, unlike other avisaurids where this face is comparatively flatter. The distal portion of metatarsal II and its trochlea deflects medially as in *A. archibaldi*, *Gettyia*, and *Mirarce* [12, 16]. The shaft of metatarsal IV is cranioplantarly compressed similar to *A. archibaldi* and *Gettyia*. The distal portion of the metatarsal gently curves laterally just proximal to the mediolaterally expanded trochlea more so than *Soroavisaurus* and *Mirarce* [12, 16, 18].

On the craniomedial surface of metatarsal II there is a large and rugose tubercle for the attachment of the *m. tibialis cranialis* (Fig 1A and 1C). This structure is elongate, beginning 29% of the way down metatarsal II and extending for 15% of its total length, similar to *A. archibaldi* both in size and position [12]. Distally, there is a weakly-developed intertrochlear

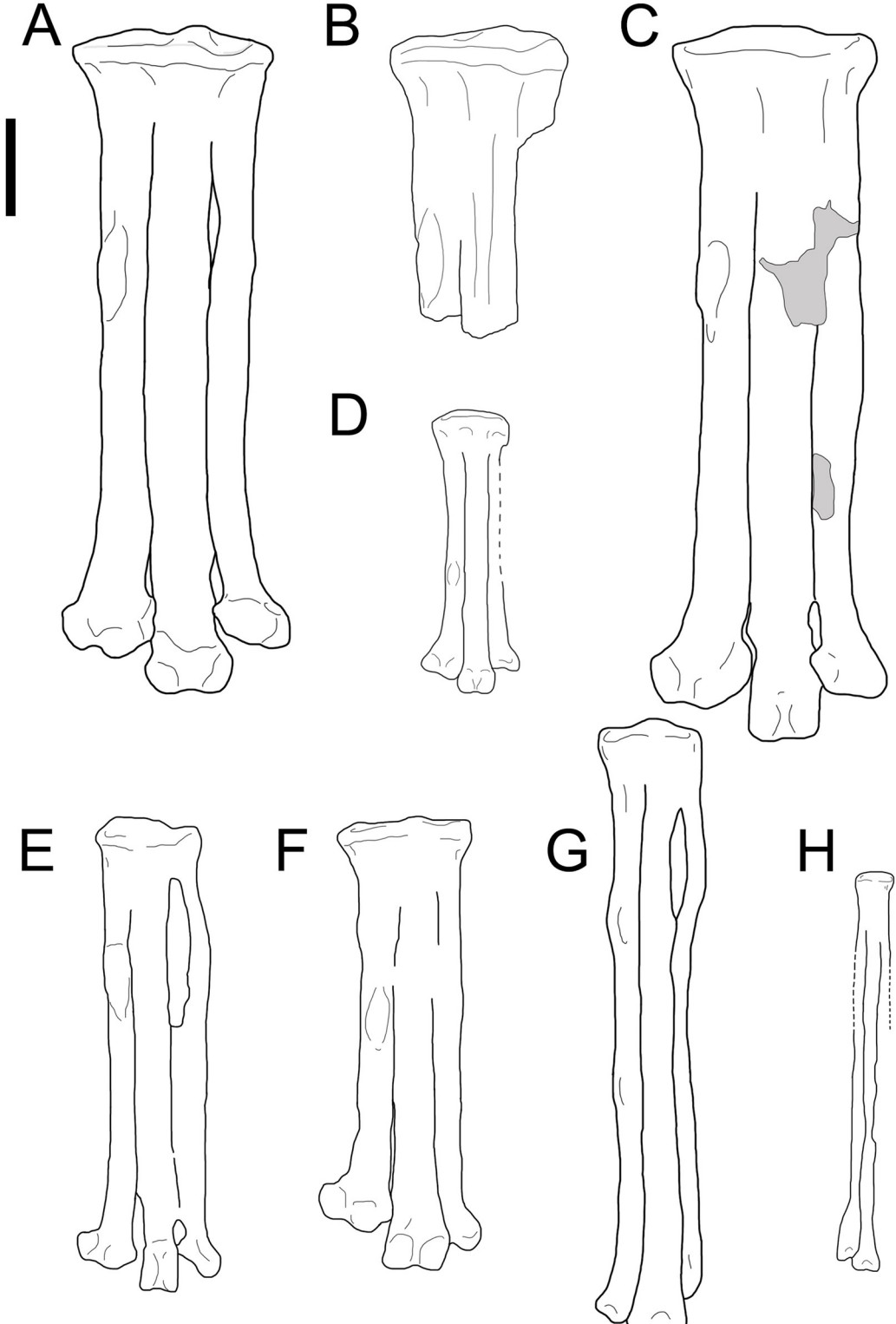

**Fig 4. Tarsometatarsus size variation among the Avisauridae and close relatives.** A) *Avisaurus darwini*, B) MOR3070, C) *Avisaurus archibaldi*, D) *Gettyia*, E) *Soroavisaurus*, F) *Mirarce*, G) *Intiornis*, H) *Neuquenornis*. All to scale. Scale bar 10 mm.

incisure between metatarsals II and III. The space extends for approximately 11% that of the total length of metatarsal III. The distal vascular foramen between trochlea III and IV begins 78% down the length of metatarsal III and terminates at approximately 86%. The mediolateral width of this space measures approximately 34% that of the midpoint width of metatarsal IV.

The laterally-projecting flange of the lateral trochlea of metatarsal II expands such that the trochlea along the plantar margin is wider than that of metatarsal III, unlike *Soroavisaurus*. In *A. archibaldi*, this flange is proportionally smaller, projects lateroplantarly, and does not fully close the gap between the trochlea of metatarsal II and III [12]. In *Mirarce*, this feature is also present, but is located more distally on metatarsal II and expands distolaterally [16]. In *A. darwini*, this distal laterally-projecting flange contacts that distomedial margin of metatarsal III, closing the gap between the trochlea of metatarsals II and III. This closure creates a condition analogous to the distal vascular foramen between the trochlea of metatarsal III and IV [16].

The medial and lateral condyles of the metatarsal III trochlea terminate approximately at the same level, whereas the medial margin extends farther distally in *A. archibaldi* [12]. As in *A. archibaldi* and *Mirarce*, the medial condyle of the trochlea of metatarsal III exhibits greater plantar projection than the lateral condyle [12, 16]. The trochlea of both metatarsals II and III are ginglymous, and exhibit well-developed, excavated, intercondylar grooves. These well-developed grooves are also present in *A. archibaldi*, *Mirarce*, *Gettyia*, and *Soroavisaurus* [12, 16, 18]. The trochlea of metatarsal IV has a reduced medial condyle such that it appears as a single, plantomedially excavated condyle, similar to other enantiornithines (e.g., *A. archibaldi*, *Avimaia*, *Imparavis*, *Mirarce)* [12, 16, 20, 31]. In distal view, the trochlea of metatarsal IV is C-shaped (medially concave), and projects farther plantarly than even the medial condyle of the metatarsal III trochlea, as in other avisaurids (e.g., *Mirarce*, *A. archibaldi*) [12, 16].

## Specimen MOR 3070—*Avisaurus* sp.

The metatarsals are all fused proximally to each other and the distal tarsals (Fig 2). Similar to other *Avisaurus*, the proximal articular surface is oval with the lateral cotyle slightly elevated relative to the medial cotyle. The proximal labrum is well-developed as in *A. archibaldi*, *A. darwini*, and *Mirarce* [12, 16]. However, unlike *A. darwini*, and *Mirarce*, MOR 3070 exhibits greater lateral to medial tapering of the labrum thickness along the dorsal margin (Fig 2A; [16]). In plantar aspect, the labrum is dorsoventrally thickest at its midpoint (i.e., above the midline of metatarsal III). Characteristic of the genus *Avisaurus*, two proximoplantar fossae bordering metatarsal III are present. These fossae are not as deep as in *A. darwini*.

Though incomplete, MOR 3070 would have had a low aspect ratio of the tarsometatarsus (~ 1:4) like other avisaurids (e.g., *Mirarce*, *Gettyia*, and *Avisaurus*). Proximally, metatarsal II is the mediolaterally widest metatarsal, with the width of metatarsal III measuring approximately 87% that of the metatarsal II. This in contrast to *Mirarce*, *A. archibaldi*, and *A. darwini*, where even proximally the width of metatarsal II is less than that of III [16]. Similar to members of *Avisaurus*, MOR 3070 exhibits a well-developed *m. tibialis cranialis* tubercle which if scaled to *Avisaurus* would place it approximately 30% down the length of metatarsal II. In contrast to other avisaurids, the *m. tibialis cranialis* tubercle of MOR 3070 projects more dorsally as opposed to dorsomedially. The disarticulated distal portion of metatarsal II exhibits similarities to *A. archibaldi* in that it exhibits a lateroplantarly-projecting flange [12]. This is disparate from the laterally-projecting morphology exhibited in *A. darwini*. The relationship of this flange with the trochlea metatarsal III is unknown, but similar to *A. archibaldi*, it likely did not contact metatarsal III (this flange contacts the trochlea of metatarsal III in *A. darwini*). Beginning prominently below the labrum shelf, a well-developed dorsally-projecting ridge extends distally down the remaining portion of the dorsal surface of metatarsal III (forming a

triangular cross section in plantar aspect). This feature is absent in *A. darwini*, and weakly developed in *A. archibaldi*.

CCM V2019.5.1 –Holotype of *Magnusavis ekalakaenis*

urn:lsid:zoobank.org:act:4A5DF7A9-06FD-45A0-B7CA-726B1936D7DA

The holotype of *M. ekalakaenis* consists of partial metatarsals II and III and a single phalanx (Fig 3). Unlike avisaurids, both preserved metatarsals exhibit subtle distal expansion. The proximal-most mediolateral width of metatarsal III measures 76% that of the distal-most portion (proximal to the trochlea) (*A. darwini*, 100%, *A. archibaldi*, 96%). Metatarsal III is both proximodistally longer and mediolaterally wider than metatarsal II along its length, typical of most enantiornithines [4].

The shaft of metatarsal II is rounded and smooth, in contrast to the well-developed peaked ridge in MOR 3070 and *A. darwini*. The distomedial deflection of metatarsal II is weakly-developed compared to the condition seen in avisaurids, exhibiting a nearly closed intertrochlear incisure between metatarsal II and III. Though not as pronounced or as large as the condition seen in avisaurids, the *m. tibialis cranialis* tubercle is well developed when compared to most enantiornithines [12, 16]. Similar to metatarsal II, the dorsal surface of metatarsal III is only weakly convex, in contrast to the strongly convex morphology in avisaurids (e.g., *Mirarce*, *A. archibaldi*, MOR 3070, and, *A. darwini*).

The condyles of the metatarsal III trochlea terminate at the same level distally. Like avisaurids, the intercondylar grooves are well-developed in both metatarsals II and III, particularly in III. The medial condyle of the metatarsal III trochlea exhibits greater dorsal and plantar projection relative to the lateral condyle. The plantar supratrochanter fossa is also well developed, and clearly demarcated lateral and medially. The mediolateral width of this depression measures approximately 75% the total width of the trochlea of metatarsal III. The lateral fossa of metatarsals II and III are well-developed.

Though digit assignment of the phalanx is uncertain, it resembles the first phalanx of pedal digit II in *Mirarce*, albeit much more dorsoventrally deep (lateral aspect ratio *Magnusavis* 1:3, *Mirarce* 1:6). The aspect ratio of the phalanx is approximately 1:3, in contrast to that seen in Mirarce (~ 1:5).

## Mass estimates

Based on the maximum diameter of the glenoid facet of the coracoid and the length of the humerus in *Mirarce*, we calculated an estimated mass of ~ 540 g (range 432–808 g) (S2 Table). Similarly, the resulting estimated body mass for *A. darwini* (DDM 1577.730) was ~ 1200 g (range 914–1712 g) and ~ 1700 g for *A. archibaldi* (range 1488–2283 g) (S3 Fig and S2 Table).

## Phylogenetic analysis results

Both the Nelson strict consensus tree and the 50% majority rule tree place all three new taxa within enantiornithines, with *A. darwini* and MOR 3070 (*Avisaurus* sp.) resolved as part of the Avisauridae clade (Fig 5).

Within the Avisauridae, strict tree consensus suggests *A. archibaldi* and *A. darwini* form a more exclusive clade supported by a metatarsal II that is approximately equal in distal extent to metatarsal IV [Character 236 1 → 0]. Additionally, *Mirarce* and *Gettyia* form a clade supported by the absence of a plantarly excavated tarsometatarsus [Character 230 1 → 0], a *m. tibialis cranialis* tubercle located near or distal to the midpoint [Character 247 1 → 2], and the absence of proximoplantar fossae [Character 254 1 → 0]. In the 50% majority tree, these two clades form a polytomy with MOR 3070 and *Soroavisaurus*. With the addition of MOR 3070 and *Soroavisaurus*, the Avisauridae clade is formed by six OTU (operational taxonomic units)

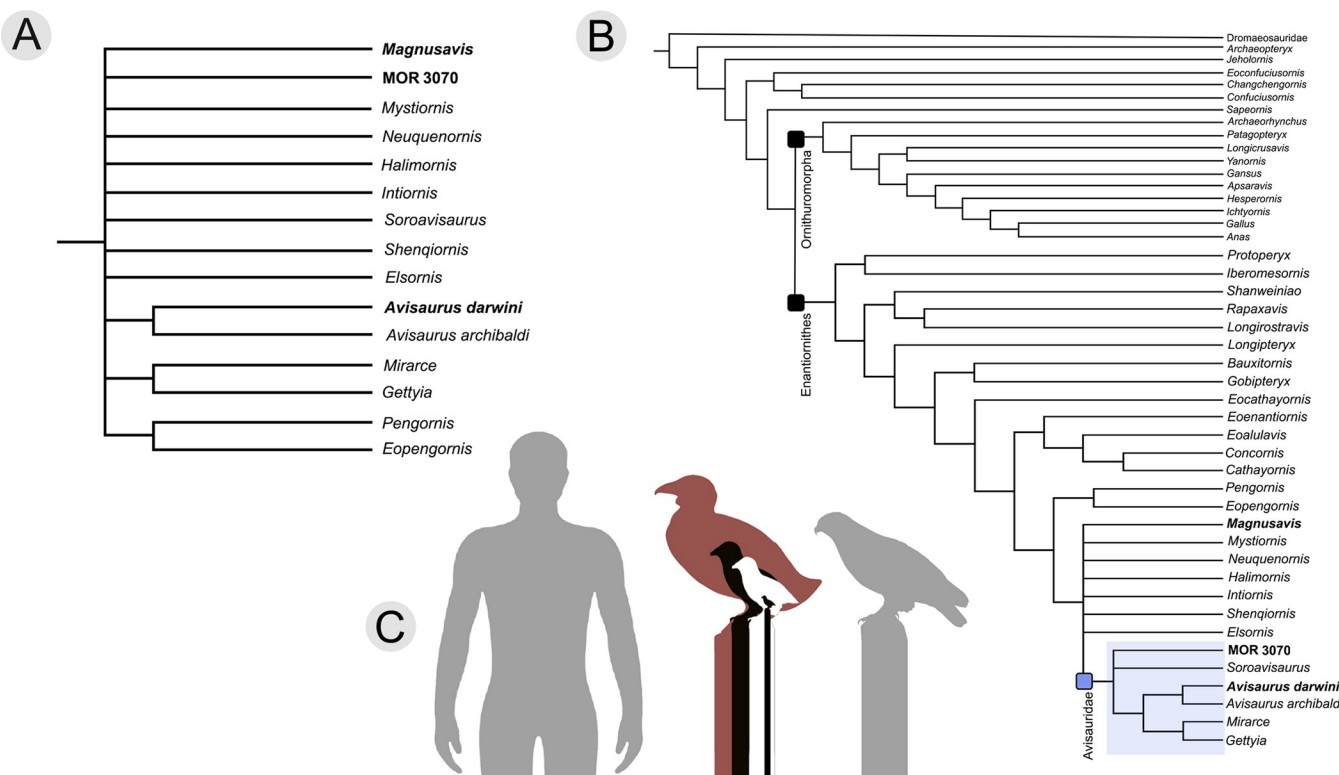

**Fig 5. Phylogenetic placement of new Hell Creek enantiornithines based on cladistic analysis.** A) A subset of the strict consensus tree focusing on taxa around the Avisauridae, and B) the full 50% majority tree. In the majority tree, the newly diagnosed Avisauridae family is comprised of six taxa, two of which are described in this publication. C) A 1.8 m tall human to scale with *A. darwini* (maroon), the largest known Early Cretaceous enantiornithine *Pengornis* (black), a medium-sized early enantiornithine, *Imparavis* (white), and finally, a minuscule enantiornithine, *Elektorornis* (smaller inset black). The extant *Buteo jamaicensis* (Red-tailed Hawk) is shown in grey.

and collectively supported by two morphologies: tarsometatarsus whose midpoint width measures at least 20% that of its length (i.e., low aspect ratio) [Character 253 1 → 0]; metatarsal II trochlea that is broader than that of metatarsal III [Character 238 0 → 1].

*Magnusavis*, *Neuquenornis*, and *Intiornis* are part of a larger polytomy outside of the Avisauridae along with *Mystiornis*, *Halimornis*, *Shenqiornis*, and *Elsornis*. With the removal of two previously identified avisaurids (*Neuquenornis*, and *Intiornis*), this family is now primarily defined not by the convex dorsal surface of metatarsal III or plantar excavation of the tarsometatarsus, characters that are both widespread in enantiornithines (e.g., *Imparavis*, *Elektorornis*, *Qiliania*) [20, 32], but instead by a proportionately robust tarsometatarsus (mediolateral width at the midpoint at least 20% that of the total proximodistal length). The removal of these two taxa differs from previous phylogenetic results which place them alongside avisaurids based on a plantar projection of the medial trochlea of metatarsal III [16]. These results support interpretations that MOR 3070 is an avisaurid and likely a member of the genus *Avisaurus* (e.g., robust morphology, two proximoplantar fossae), and one that differs from other avisaurids; however, in light of the fragmentary remains available at present, we refrain from naming this specimen as a new taxon.

## Discussion

### Size diversity in late cretaceous enantiornithines

Here we describe three new large birds from the Maastrichtian Hell Creek Formation in Montana, all of which are placed in the clade Enantiornithes (Fig 5). These three new

enantiornithines not only increase the known taxonomic diversity of the Hell Creek avifauna, but they also add to existing evidence that members of this speciose clade achieved proportionally larger body sizes near the end of the Cretaceous.

Most enantiornithines described from Lower Cretaceous deposits, notably the Hauterivian to Aptian Lagerstätte in Spain and China (~ 130–120 Ma), and Upper Cretaceous amber specimens from Myanmar (~ 99–100 Ma), are similar in size to sparrows (families Passeridae, Emberizidae) within the range of ~ 10–70 grams and thrushes (family Turdidae) within the range of ~ 20–170 grams using the length of the humerus as a means of comparison [4, 20, 33–38]. Currently, the smallest Upper Cretaceous enantiornithines are Burmese amber specimens (*Elektorornis*, predicted to the size of 3–5 g trochilids based on hindlimb measurements) and the largest being the holotype of *Pengornis houi* which is similar in size to larger members within Psittaciformes within the range of 600–750 grams [7, 34, 39]. Mass estimates for avisaurids reveal even larger Late Cretaceous enantiornithines with *Mirarce*, *A. darwini*, and *A. archibaldi* estimated as having body masses of ~ 540 g, ~ 1200 g, and ~ 1700 g respectively (see methods and results sections).

The description of two new large-bodied avisaurids (*A. darwini*, *Avisaurus sp.*), and another proportionally large enantiornithine (*Magnusavis*), provides additional evidence in support of an observed trend in enantiornithines towards greater body size diversity over time [4, 13, 16, 33]. This increase in body size disparity towards the end Cretaceous is also observed in the Ornithuromorpha, the clade that includes Neornithes [40]. It has been suggested that larger body size among ornithuromorphs may have been a key contributing factor in surviving the end Cretaceous mass-extinction [40, 41]. Although depositional environments yielding bird fossils in Upper Cretaceous North American deposits are biased toward the preservation of larger organisms, this does not explain the apparent absence of larger enantiornithines in Lower Cretaceous Lagerstätte like the Jehol, which records larger birds like *Jeholornis* and *Sapeornis* but only small enantiornithines [42, 43] Therefore, the signal indicating increased size diversity in the Late Cretaceous is likely a genuine one (Fig 6). Fragmentary smaller-bodied enantiornithines have been recovered from the Late Cretaceous of both North and South America (Hell Creek Formation and William's quarry, respectively) [3, 40].

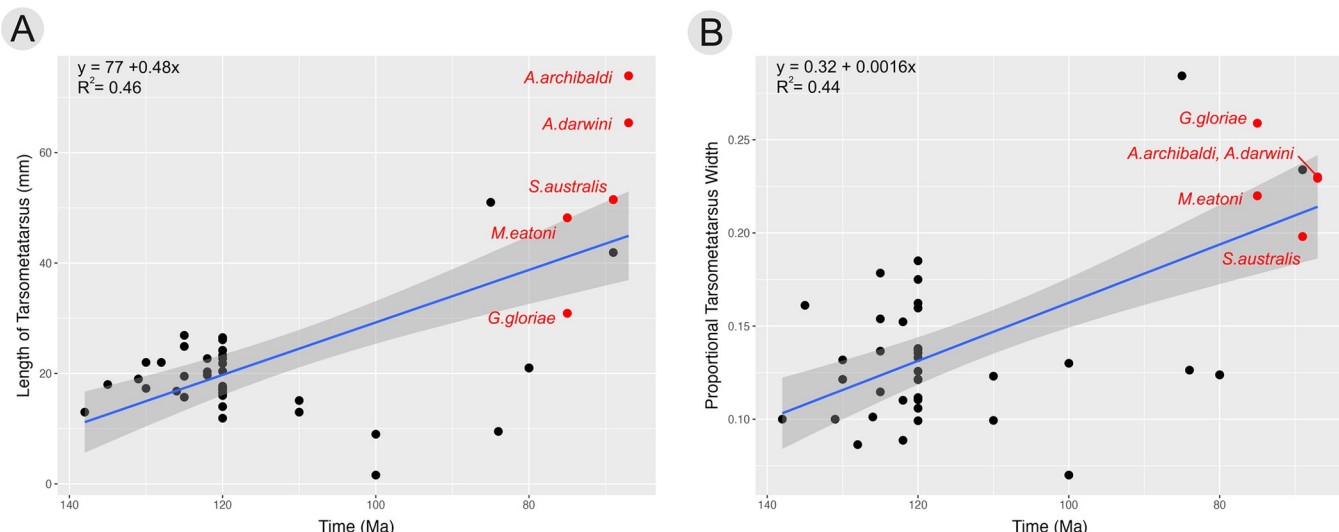

**Fig 6.** Regression analyses of A) tarsometatarsus length against time, and B) proportional tarsometatarsus mediolateral width against time. Both result in significant (p < 0.01) positive associations with an increase in body size over time. Avisaurids are colored red and all other enantiornithine are colored black.

In enantiornithines, both an increase in tarsometatarsus length and proportional mediolateral width (at the midpoint) are significantly positively associated with time (p < 0.01) (Fig 6). Though well-preserved enantiornithine tarsometatarsal material from 119–80 Ma is limited, represented by two specimens from Myanmar preserved in amber (e.g., DIP V 19354, *Elektorornis*) [32, 44], before 119 Ma the average tarsometatarsus length was 19 mm and the proportional mediolateral thickness was 12% (~ 2.28 mm) (aspect ratio of ~ 1:8). In contrast, these averages are 43 mm and 21% (mediolateral thickness of 9 mm) (aspect ratio 1:5) after ~ 80 Ma (Fig 6). This pattern is likely exaggerated by the poor Upper Cretaceous fossil record and the fact no postcranial material has yet been described from William's Quarry, although it is reportedly present in abundance [3].

## Functional morphology of the avisaurid tarsometatarsus

The Avisauridae are known almost entirely from tarsometatarsi. Members of the Avisauridae exhibit a combination of morphologies of the tarsometatarsus that are distinct from other enantiornithines and suggest raptorial behaviors including: the aspect ratio of the tarsometatarsus; the morphology and location of the *m. tibialis cranialis* tubercle; and the well-developed grooves of the trochleae. Specifically, these characteristics in avisaurids show striking similarities to strigids (owls, order Strigiformes) and accipitrids (hawks, eagles, order Accipitriformes). However, the differences between avisaurids and these extant clades indicates that neither provides a direct analogy for the ecological role occupied by avisaurids in the latest Cretaceous. The unique union of morphologies observed in avisaurids demonstrates that they likely occupied a predatorial ecological niche morphologically unrepresented in extant forms.

## Aspect ratio of the tarsometatarsus

The morphology and topography of the tarsometatarsus greatly varies among extant birds, and can signal certain ecological behaviors such as wading, cursoriality, perching, prey grasping abilities, and pedal digit to substrate interaction [45–51]. When assessing the length and width of the tarsometatarsus of Mesozoic taxa, avisaurids overlap with a range of terrestrial nonvolant theropods (Fig 7 and S4 Table). Thus, it is not surprising avisaurids were originally

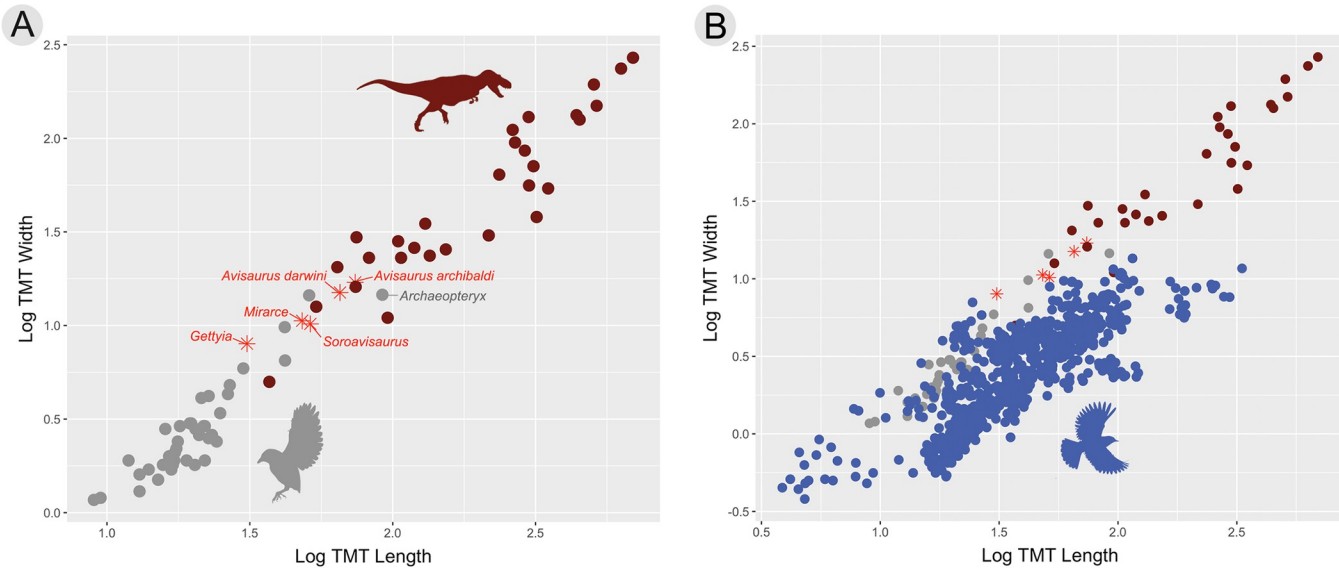

**Fig 7.** Assessing aspect ratio of A) extinct avian and non-avian theropods, and then with the addition of B) extant volant birds. The grey dots represent extinct volant theropod genera. The red dots represent non-avian theropod genera. Blue dots represent extant avian genera. Refer to S4 Table for data.

interpreted as non-avian, or as having reduced flight capabilities [12]. However, when modern birds are added to the dataset, avisaurids cluster with larger birds like pelicans, owls, and hornbills weakening support for using the aspect ratio of the tarsometatarsus alone to predict volancy (Fig 7).

Small bodied strigids (< 350 g) exhibit tarsometatarsi aspect ratios very similar to avisaurids (e.g., *Surnia ulula*, 1:4, *Aegolius funereus*, 1:5, *Glaucidium brasiliensis* 1:5) (Fig 8). Similarities such as the aspect ratio of the tarsometatarsus, location of the *m. tibialis cranialis* tubercle, and the deeply notched trochlea of the metatarsal all suggest extant raptorial birds, and more specifically strigids, may be the best functional analogues for understanding hindlimb function in avisaurids. Strigids are able to produce greater grip (i.e., constriction of pedal digits) force than other similar sized birds of prey [48, 52, 53]. The proportionally more elongate tarsometatarsi of accipitrids (aspect ratios of 1:20–1:10) are better suited for rapid movements (i.e., grabbing, snatching), whereas the proportionally stouter tarsometatarsi of strigids are better equipped for forceful constriction [48, 53]. By lowering the aspect ratio of the tarsometatarsus relative to their tibiotarsus, strigids reduce the length between the fulcrum (ankle joint), moment of action (muscle tendon insertion), and the resistance force (the prey), therefore reducing the required muscular force (i.e., torque) to lift or grip prey compared to longer-legged accipitrids [54]. In combination with having more mechanically efficient legs for producing dorsiflexing torque, strigids also possess mediolaterally wider muscle tendons and proportionally larger flexor muscles (e.g., *m. flexor digitorum longus*) than similarly sized accipitrids [48, 49, 52, 53]. The lower aspect ratio of the tarsometatarsus accommodates these wide muscle tendons, resulting in more applied force (i.e., contraction) over a given area. The *m. flexor digitorum longus*, which is responsible for contracting pedal digits II-IV, is the largest observable lower hindlimb muscle in strigids; the muscle belly, located on the caudal surface tibiotarsus, is so large that it also protrudes around the medial and lateral borders of the tibiotarsus [53, 54]. The combination of potentially mediolaterally wider tendon insertions on the tarsometatarsus and greater force from increased flexor muscle mass in avisaurids may have resulted in comparatively more powerful and efficient pedal grip relative to other enantiornithines.

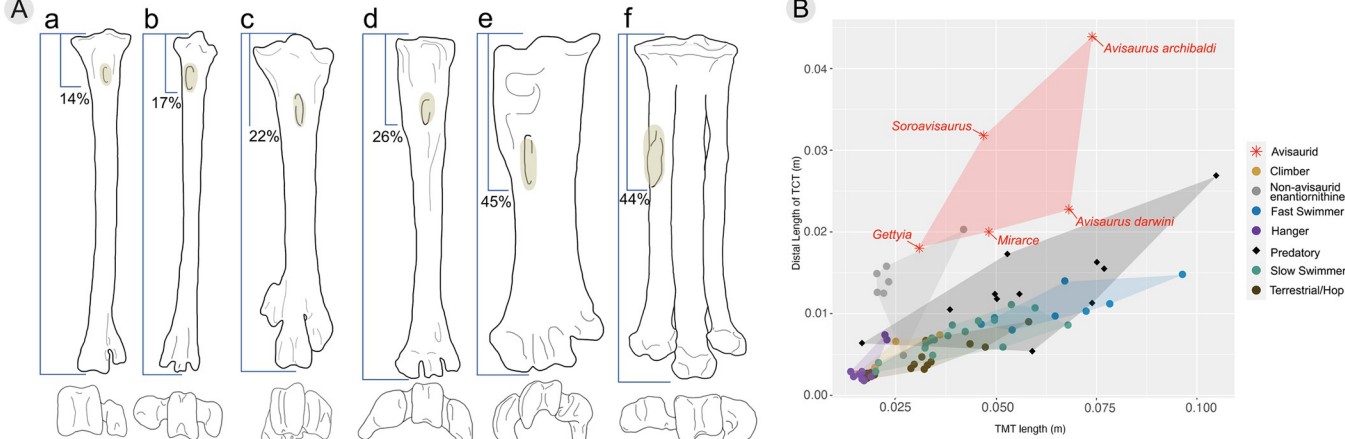

**Fig 8.** A) The left tarsometatarsi in dorsal and distal aspects with the *m. tibialis cranialis* tubercle highlighted in a) *Struthio camelus* (terrestrial ratite), b) *Turdus olivator* (arboreal passerine), c) *Cygnus buccinator* (semi-aquatic anseriform), d) *Buteo jamaicensis* (raptorial accipitrid), e) *Surnia ulula* (raptorial strigid), f) *Avisaurus darwini*. The distal view of each taxon is represented underneath the dorsal view to show variations of the intertrochlear grooves. Tarsometatarsi are not to scale. Reference photos of tarsometatarsi sourced from morphosource. B) Tarsometatarsus length (in meters) against the proportional distal location of the *m. tibialis cranialis* tubercle (TCT) in extant birds and enantiornithines, including avisaurids (marked as asterisks). Extant birds are divided into their primary means of locomotion, following data from Zeffer and Norberg [52].

Similarly, the tarsometatarsi of avisaurids exhibit lower aspects ratios than all other enantiornithines (with the exception of *Bauxitornis*), suggesting that members within this clade had analogously powerful flexion of pedal digits II-IV [55]. However, strigids are zygodactyl (with digits I and IV reversed), whereas avisaurids and accipitrids are anisodactyl (only digit I reversed). In contrast to strigids, accipitrids exhibit greater developed *m. flexor hallucis longus* tendons [49]. This facilitates greater force to the contraction of the hallux (and partially to digit II) which is necessary as it opposes the combined forces generated by the three forward facing digits (II-IV) [49, 53]. The mediolaterally wide and well-developed plantar excavation of the tarsometatarsus in large avisaurids (e.g., *Mirarce*, *Gettyia*, *Avisaurus*) is a space to accommodate mediolaterally expanded muscle tendons (of the *m. flexor hallucis longus* and *m. flexor digitorum longus*) as well as presence of a large *m. flexor hallucis brevis* belly (originating on the proximal part of the plantar surface), facilitating greater contractile grip. Avisaurids appear to unite the pedal arrangement of accipitrids and the mechanical benefits of the low aspect ratio tarsometatarsi in strigids.

Additionally, the well-developed, shelf-like labrum along the proximoplantar surface suggests the presence of an in-vivo cartilaginous hypotarsus, similar to that described in *Confuciusornis* [56]. If present, this feature would both aid in supporting the tendons of the digital flexor muscles and function as a moment arm for the ankle joint, allowing for greater tendon stability and ventroflexion of the lower hindlimb, respectively [57, 58].

### Functional significance of the *m. tibialis cranialis* tubercle

Present in confuciusornithiforms, ornithuromorphs, and enantiornithines, the tubercle on metatarsal II has been hypothesized to represent the insertion point for the *m. tibialis cranialis* [4]. This feature is particularly pronounced (dorsally projecting) in enantiornithines, present in nearly all taxa in which preservation allows observation. Interestingly, this feature is noticeably absent in non-avian theropods, and located on metatarsal III in a select number of Mesozoic birds (e.g., *Mystiornis*, *Confuciusornis*, *Ichthyornis*) [59]. In extant birds, the *m. tibialis cranialis* originates on the proximocranial surface of the tibiotarsus and inserts on the dorsoproximal surface of the tarsometatarsus [50, 60]. As the muscle tendon dorsodistally extends from the distal tibiotarsus to the tarsometatarsus, it is secured in place by a retinaculum on the craniodistal surface of the tibiotarsus, just proximal to the ankle joint [50, 60]. Contraction of this muscle dorsiflexes the ankle, bringing the dorsal face of the tarsometatarsus towards the cranial surface of the tibiotarsus [50, 60]. The larger the insertion point (i.e., tubercle), the larger the corresponding muscle tendon, resulting in more powerful dorsiflexion. Likewise, as the insertion point migrates distally, it increases the in-lever of the system while the out-lever remains the same, increasing the system's mechanical advantage and ultimately allowing for increased dorsiflexion while carrying weight in the pes [54, 60, 61] (Fig 9).

Variation in the location of this muscle attachment among extant birds makes it a useful osteological correlate for determining interaction between substrate (or prey) and the pedes [50, 60]. Dorsiflexion is necessary in maintaining balance by lowering the center of mass when perching, which is why this feature is often more well-developed and proximally located in passerines than in terrestrial birds [50], and similarly why this feature is well-developed in arboreal enantiornithines (Fig 8). Extant birds that use their legs for swimming have even more prominent *m. tibialis cranialis* tubercles that are more distally located than perching birds. This facilitates dorsiflexion of the foot against the resistance of water, to subsequently extend and produce thrust (i.e., paddling). However, raptorial birds exhibit both the largest and most distally-located *m. tibialis cranialis* tubercles [50] (Fig 8). Similar to the relationship between raptorial and non-raptorial birds, avisaurid *m. tibialis cranialis* tubercles differ from other enantiornithines in being larger and more distally located (Fig 8 and S3 Table).

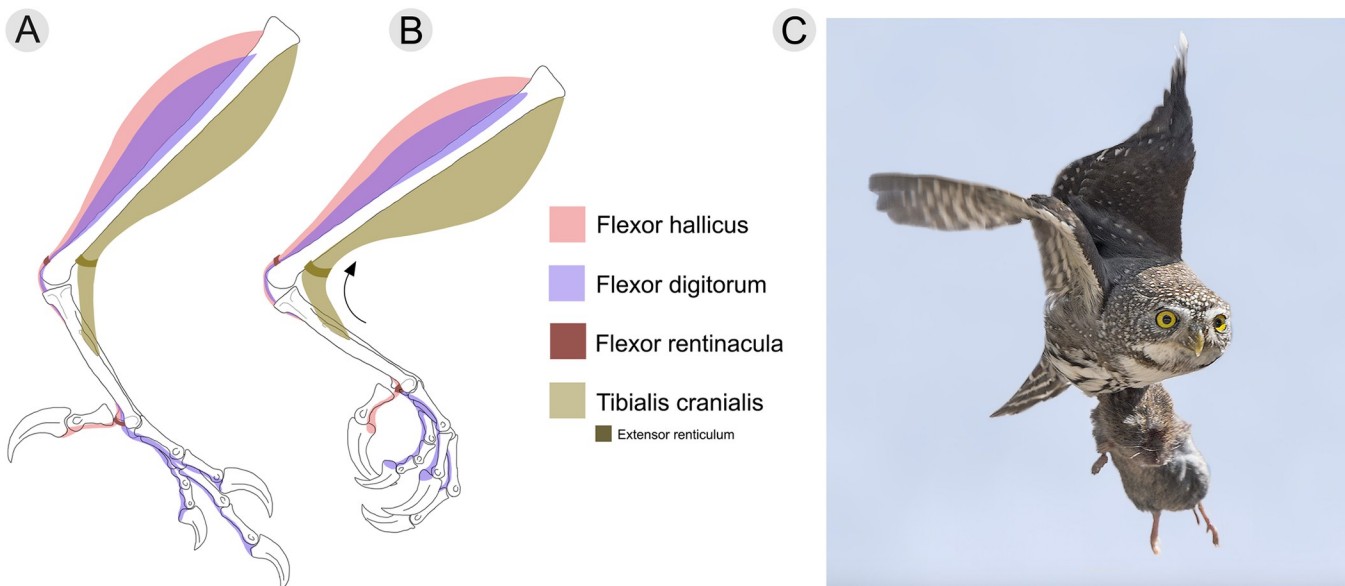

**Fig 9.** A) Reconstruction of an articulated left tibiotarsus, tarsometatarsus, and pes of *Avisaurus darwini* with muscular reconstruction of the primary pedal flexors (caudal surface) and *m. tibialis cranialis* (cranial surface) in medial aspect. A) ankle joint shown in neutral position. B) ankle joint shown in dorsiflexion. Pedal elements are based on *Mirarce* which preserves nearly all pedal phalanges (Atterholt et al., 2018). The tibiotarsus is based on the distal portion preserved in *Mirarce* and the reconstruction given in Atterholt et al. [16]. Muscles such as the gastrocnemius and extensors are not included to more clearly show the two major flexor muscles and the *m. tibialis cranialis*. The tibialis cranialis muscle attaches to the hypertrophied tubercle of metatarsal II. The action of the tibialis cranialis is flexed to dorsiflex the ankle and invert the tarsometatarsus. This feature is hypertrophied in birds of prey compared to other extant birds, and is associated with an increased capacity to grasp prey with the pes. C) A member of the genus *Glaucidium* exhibiting the carrying strength that comes from having an enlarged and distally-migrated *m. tibialis cranialis* tubercle. Within this taxon, prey items may exceed three times the body mass of the owl [62, 63]. Photo used with permission and credited to Roy Priest.

Variation in both the size and distal location of this muscle attachment among raptorial birds reflects differences in acquisition and proportional size of prey [49, 50]. Along with the size and distal location of the tubercle, the relative medial location on the tarsometatarsus also affects the degree of pedal inversion that occurs when the *m. tibialis cranialis* is contracted (Fig 10). Inversion of the pes while grasping prey increases grip stability as the force being generated pushes against the opposing foot. The more medial the location of the tubercle, the greater the degree of pedal inversion [49, 50]. Compared to accipitrids and falconids, strigids also have the most distomedially located *m. tibialis cranialis* tubercles. In particular, smaller bodied owls, like *Glaucidium* tend to exhibit greater distomedial migration of the attachment compared to larger strigids, further aiding in supporting proportionally larger prey when in flight. For example, *Glaucidium gnoma*, though diminutive (up to ~ 80 g) will subdue and travel with prey items that often rival or exceed their own body mass (e.g., *Microtus* sp., 60 g; *Coccothraustes* sp., 60 g; *Callipepla sp.*, 160 g) [50, 63] (Fig 9B).

The *m. tibialis cranialis* tubercle in avisaurids is larger and more distomedially located on the tarsometatarsus than in any Mesozoic bird, non-avian theropod, and nearly every extant bird [12]. The presence of the large distomedially located *m. tibialis cranialis* tubercle in avisaurids, together with the lower aspect ratio of the tarsometatarsus, suggests a functional similarity with extant birds of prey that distinguishes them from other enantiornithines. Based on the numerous shared morphologies with strigids, avisaurid tarsometatarsi were likely adapted for strong dorsiflexion and carrying proportionally large prey items while volant (Fig 11). However, it should be noted that the largest avisaurids are estimated to be nearly twenty times larger than *G. gnoma*, (*G. gnoma* 80 g, *A. archibaldi* ~ 1700 g) which limits the proportional size of prey that could likely be lifted given that as an object's surface area increases, its volume

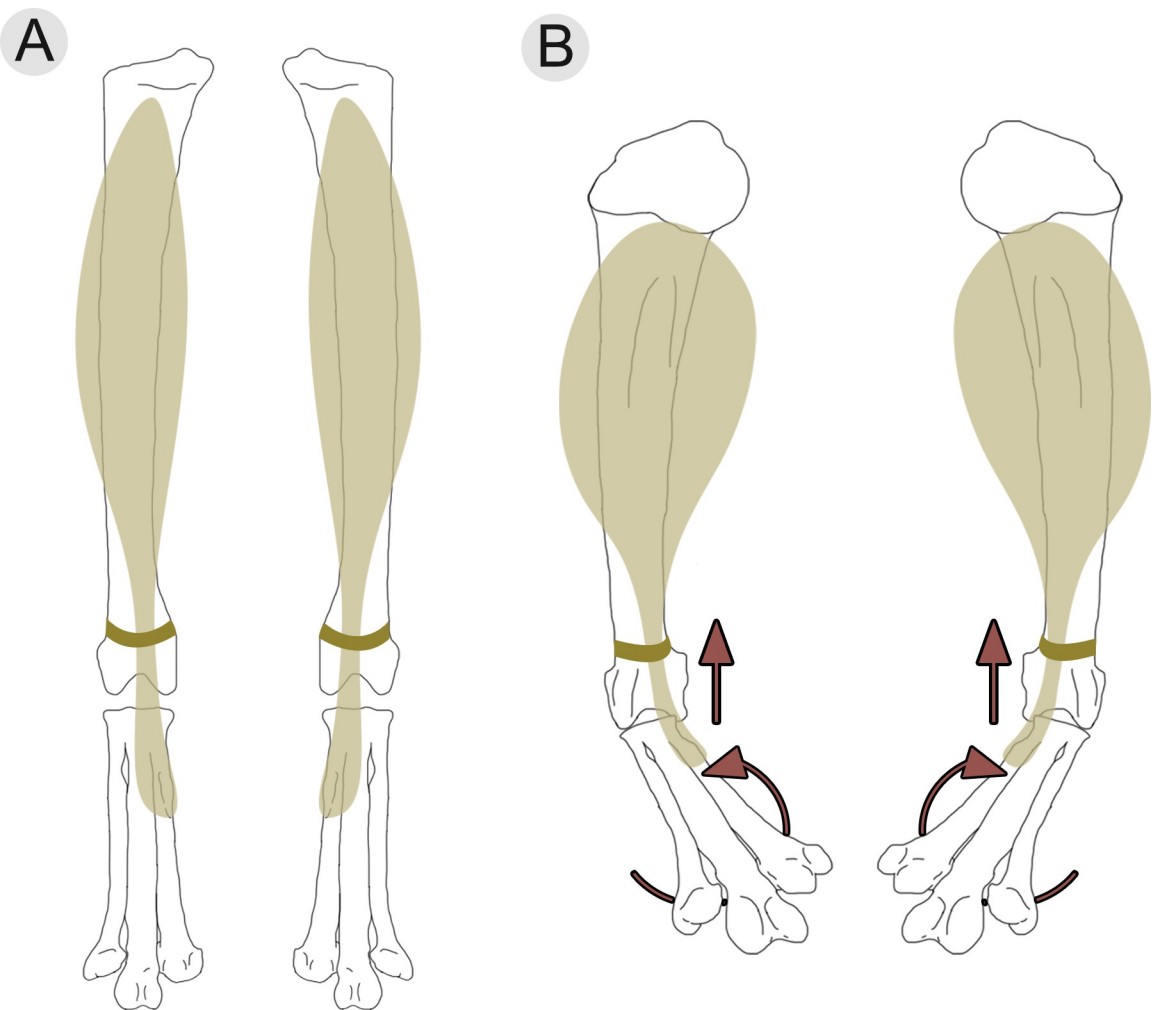

**Fig 10.** Action of the tibialis cranialis muscle on the ankle joint, shown in a A) resting position and a B) contracted position. As the muscle contracts, not only does the ankle dorsiflex, bringing the tibiotarsus and tarsometatarsus closer together, but the tarsometatarsus and phalanges invert (i.e., elevation of the medial border of the foot).

increases at a greater rate (i.e., the square-cube law). Carrying similar proportionally-sized prey as some strigids (e.g., *Surnia*, *Glaucidium*, *Aegolius*) may have been more plausible for smaller-bodied avisaurids like *Gettyia* (Zeffer and Norberg, 200). *Gettyia* (Fig 4D) exhibits the proportionally distal-most *m. tibialis cranialis* tubercle. It is also the smallest known avisaurid (~ 540 g), with a tarsometatarsal length that is less than half that of *A. archibaldi* (30.9 mm, *A. archibaldi* 73.9 mm) [16] (Fig 8A).

### Significance of trochlear groove development

Birds which rely on increased constriction of prey (e.g., birds of prey) exhibit well-developed and deeply-grooved metatarsal trochleae, which correspond to both proportionally long and robust pedal digits [47, 53, 64, 65] (Fig 8). This grooved morphology exhibits distinctly developed excavations between the medial and lateral condyles. The deeply-grooved metatarsal trochlea corresponds to phalanges with greater-projecting intercotylar crests, and excavated articular cotyla. This morphology results in a larger articular surface between the proximal phalanges and the tarsometatarsus, resulting in a restricted range of motion at the

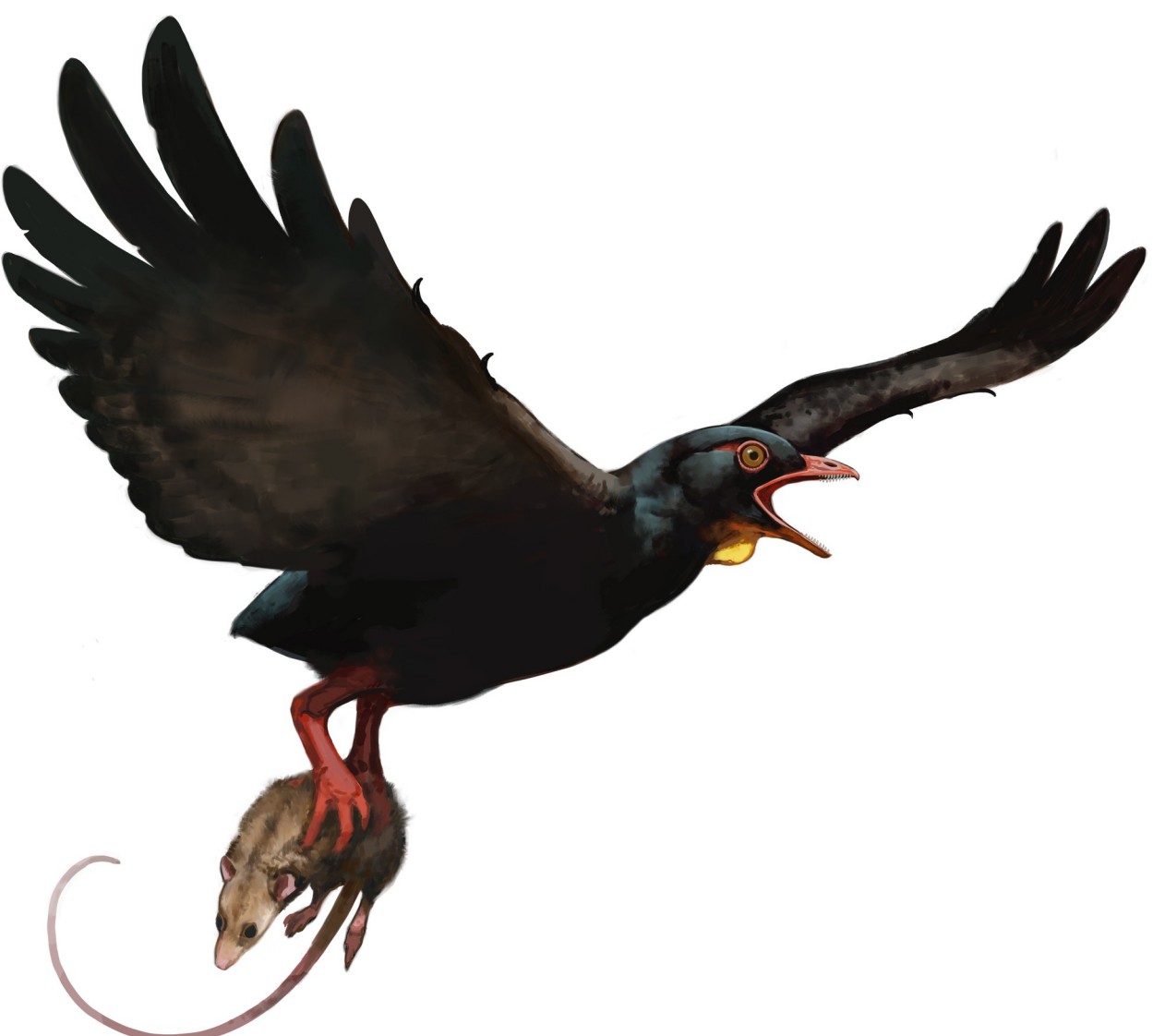

**Fig 11. Reconstruction of an avisaurid (e.g., *A. darwini*).** Morphology of the tarsometatarsus suggests that these large birds engaged in raptorial behavior and could carry proportionally large prey. Illustration done by Ville Sinkkonen.

metatarsophalangeal joint when the flexor tendons contract, (i.e., gripping an object) [47, 48, 66]. Pedal digits II-IV of strigids have mediolaterally wider tendons of the *m. flexor digitorum* than in accipitrids, facilitating stronger contraction and therefore grip strength [47, 66]. Together with the low aspect ratio of the tarsometatarsus, the more robust pedal digits of strigids are suggested to also aid in constricting prey [47, 52, 61, 66, 67]. This may explain why strigiforms have been found to apply more constrictive force than is necessary to kill their prey items [47, 66] Well-developed (compared to other enantiornithines) trochlear grooves are present in the mediolaterally wide trochleae of metatarsals II and III of *A. archibaldi*, *A. darwini*, *Mirarce*, *Soroavisaurus*, and *Gettyia*, suggesting similarly robust and powerful pedal digits, reminiscent of those seen in extant birds of prey [12, 16, 47, 48, 64]. Together with the absolute larger size of avisaurids compared to other enantiornithines, a proportionally high level of applied pedal digit pressure (i.e., constriction) is plausible for members of this clade.

## Conclusion

Here we describe three new enantiornithines, two of which are new avisaurids, all from exposures of the latest Cretaceous Hell Creek Formation in Montana. Both new avisaurids are larger than the largest Early Cretaceous enantiornithine. This supports the apparent trend towards increased body mass diversity in the Late Cretaceous among enantiornithines with large-bodied forms being most heavily represented in the Late Cretaceous fossil record. By increasing body size, this unique bird lineage explored new ecospaces suggesting that enantiornithines were still diversifying near the end Cretaceous. The morphology of the avisaurid tarsometatarsus suggests some Late Cretaceous enantiornithines adopted raptorial ecologies. Avisaurid tarsometatarsi exhibit low aspect ratios, proportionally large and distomedially located *m. tibialis cranialis* tubercles, and well-developed grooved metatarsal trochleae. All of these morphologies are uniquely found in members of extant raptorial birds, specifically strigids and accipitrids, and suggest similar powerful pedal constriction behaviors in an anisodactyl pedal formation. By combining multiple aspects of the tarsometatarsus of both extant accipitrids and strigids, avisaurids represent a group of enantiornithines which employed raptorial behaviors that lack modern morphological analogues.

## Supporting information

**S1 Fig. Analysis of 863 extant birds (317 species) measuring mass in grams against length of the tarsometatarsus.** Data from Field et al., (2013).
(TIF)

**S2 Fig.** Close-up photos of the proximoplantar fossa on the plantar face of (A) *Avisaurus darwini*, and (B) *Avisaurus* sp. MOR 3070. The depressions are indicated by the black arrows.
(TIF)

**S3 Fig. Plotting three avisaurids to approximate mass.** Utilizing the dataset from Field et al., (2013), three avisaurids (*A. archibaldi*, *A. darwini*, *Mirarce*) were plotted against (A) all sampled extant birds, and then against all birds, (B) by family. Once closely associated families were identified, an averaged mass estimate could be calculated using data collected within Field et al., (2013).
(TIF)

**S1 Table. Sampled enantiornithine tarsometatarsi length and width at the midpoint (mm).**
(CSV)

**S2 Table. Field et al.** (2013) measurements used for weight estimations in extant birds and predictive masses for selected avisaurids based on averaged results.
(CSV)

**S3 Table. Dataset from Zeffer and Norberg (2003) with added enantiornithines to assess distal location of the *m. tibialis cranialis* tubercle.**
(CSV)

**S4 Table. Dataset used for comparative aspect ratios of the tarsometatarsus between avian and non-avian theropods.** All extant taxa used are from Field et al., 2013.
(CSV)

## Acknowledgments

Thank you to the editor and to both reviewers for their constructive suggestions for our manuscript. We would like to acknowledge the field crews of Nathan Carroll (Carter County

Museum), Thomas Carr (Dinosaur Discovery Museum), and Frankie Jackson (Montana State University). Without their effort, these new taxa would not have been found. Thank you to Daigo Yamamura for providing info on the collection of MOR 3070. Thank you to the Bureau of Land Management and Montana Fish, Wildlife & Parks for providing access to lands where these discoveries were made. Thank you to Samantha Clark who illustrated the tarsometatarsi in Figs 1–3, and to Ville Sinkkonen who illustrated Fig 11. Thank you to Akiko Shinya for assistance in additional preparation of the fossils.

## Author Contributions

**Conceptualization:** Alexander D. Clark, Jessie Atterholt, John B. Scannella, Nathan Carroll, Jingmai K. O'Connor.

**Data curation:** Alexander D. Clark.

**Formal analysis:** Alexander D. Clark.

**Investigation:** Alexander D. Clark, Jessie Atterholt, Jingmai K. O'Connor.

**Methodology:** Alexander D. Clark.

**Visualization:** Alexander D. Clark.

**Writing – original draft:** Alexander D. Clark.

**Writing – review & editing:** Jessie Atterholt, John B. Scannella, Jingmai K. O'Connor.

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
