## [Decision Letter · Decision Letter 0]

2 Aug 2024

PONE-D-24-29019New enantiornithine diversity in the Hell Creek and the functional morphology of the avisaurid tarsometatarsusPLOS ONE

Dear Dr. Clark,

Thank you for submitting your manuscript to PLOS ONE. After careful consideration, we feel that it has merit but does not fully meet PLOS ONE’s publication criteria as it currently stands. Therefore, we invite you to submit a revised version of the manuscript that addresses the points raised during the review process.

We look forward to receiving your revised manuscript.

Kind regards,

Felipe Lima Pinheiro, Ph.D

Academic Editor

PLOS ONE

Journal Requirements:

2. In your manuscript, please provide additional information regarding the specimens used in your study. Ensure that you have reported human remain specimen numbers and complete repository information, including museum name and geographic location. 

For more information on PLOS ONE's requirements for paleontology and archeology research, see https://journals.plos.org/plosone/s/submission-guidelines#loc-paleontology-and-archaeology-research.

3. Please take this opportunity to be sure you have met all of our guidelines for new species. For proper registration of a new zoological taxon, we require two specific statements to be included in your manuscript.

1) In the Results section, the globally unique identifier (GUID), currently in the form of a Life Science Identifier (LSID), should be listed under the new species name, for example:

Anochetus boltoni Fisher sp. nov. urn:lsid:zoobank.org:act:B6C072CF-1CA6-40C7-8396-534E91EF7FBB

Another LSID for the manuscript itself should also appear within the Nomenclature statement. You will need to contact Zoobank (zoobank.org/About) to obtain a GUID (LSID). You should receive one LSID for your manuscript and a separate, unique LSID for the new species. 

2) Please also insert the following text into the Methods section, in a sub-section to be called ""Nomenclatural Acts"":

The electronic edition of this article conforms to the requirements of the amended International Code of Zoological Nomenclature, and hence the new names contained herein are available under that Code from the electronic edition of this article. This published work and the nomenclatural acts it contains have been registered in ZooBank, the online registration system for the ICZN. The ZooBank LSIDs (Life Science Identifiers) can be resolved and the associated information viewed through any standard web browser by appending the LSID to the prefix ""http://zoobank.org/"". 

The LSID for this publication is: urn:lsid:zoobank.org:pub: XXXXXXX. 

The electronic edition of this work was published in a journal with an ISSN, and has been archived and is available from the following digital repositories: PubMed Central, LOCKSS [author to insert any additional repositories].

All PLOS ONE articles are deposited in PubMed Central and LOCKSS. If your institute, or those of your co-authors, has its own repository, we recommend that you also deposit the published online article there and include the name in your article.

Following a recent ruling by the International Commission on Zoological Nomenclature, electronic journals are now a valid format for publication of new zoological taxa. In order to ensure the valid publication of your new species, please be sure to include the updated version of Nomenclatural Acts (above). 

A complete explanation of our guidelines for publishing new species can be found on our website: http://www.plosone.org/static/guidelines#zoological.

5. Please amend either the title on the online submission form (via Edit Submission) or the title in the manuscript so that they are identical.

6. We are unable to open your Supporting Information file [AvisaurusTMT_Rcode (for journal).R]. Please kindly revise as necessary and re-upload.

**Additional Editor Comments:**

I'm pleased to inform you that both reviewers considered the manuscript to have high merit and be highly suitable for publication in PLOS ONE. The suggestions, I believe, will not require significant effort, but I recommend paying special attention to the comments from the first reviewer.

Felipe Pinheiro

Reviewers' comments:

Reviewer's Responses to Questions

**Comments to the Author**

1. Is the manuscript technically sound, and do the data support the conclusions?

Reviewer #1: Partly

Reviewer #2: Yes

2. Has the statistical analysis been performed appropriately and rigorously? 

Reviewer #1: Yes

Reviewer #2: Yes

3. Have the authors made all data underlying the findings in their manuscript fully available?

Reviewer #1: No

Reviewer #2: Yes

4. Is the manuscript presented in an intelligible fashion and written in standard English?

Reviewer #1: Yes

Reviewer #2: Yes

5. Review Comments to the Author

Reviewer #1: Clark et al. describe three new specimens of avisaurids birds, and use them to support the existing hypothesis that avisaurids were raptorial birds. Attached to my review is a PDF of the manuscript with comments made directly.

The descriptive portions of the manuscript are well-executed. Morphology is detailed and comparisons to establish new species are appropriate. I have requested some more detail on the Methods, but many of these requests could easily be met with a supplemental file.

However, there are some issues in the authors’ evidence presented for avisaurids as raptorial birds. In particular, the authors place much emphasis on the similarity between avisaurids and owls. In doing so, they ignore several major differences between owls and avisaurids and cite several similarities also shared with other raptorial birds. The authors mention briefly in the Abstract and Conclusion that avisaurids likely inhabited a raptorial niche not reminiscent of any extant group, and I believe this hypothesis has more support than using owls as analogues and should be more fleshed-out in the body of the manuscript.

Finally, most of the underlying data for this study has not been provided. None of the supplemental tables nor the .csv files which the uploaded R code relies on were a part of the submission (at least not a part I could access).

Overall, I believe this work is publishable with relatively minor revisions (including all supplemental files, more detailed Methods, removing commented inaccuracies from Discussion and lessening emphasis on owls as full analogues) but can be made much more impactful with more major revisions (building a more cohesive argument around Avisauridae’s unique raptorial niche).

The authors' work is both interesting and well-executed and would be an excellent addition to PLOS ONE.

Reviewer #2: The manuscript is in good shape and ready for publication. I have only a few comments:

1- I wouldn't call "Williams Quarry" a Lagerstatte. There is no evidence of soft tissue preserved or any type of other exceptional type of preservation. I believe WQ is more appropriately referred to as a 'bonebed'. You may want to also cite a recent paper on a basicranium from WQ as this paper refers to the discovery of over 1000 enantiornithine bones at this quarry (Chiappe LM, et al 2022. Proc. R. Soc. B 289: 20221398. https://doi.org/10.1098/rspb.2022.1398)

2. The passage in lines 446-448 is confusing. The authors say that "Currently, the smallest Upper Cretaceous

enantiornithines are Burmese amber specimens (Elektorornis, approximately the size of 3 g extant

trochilids) and the largest is the holotype of Pengornis houi [similar in size to large psittaciforms (600-

448 750 grams) (Zhou et al., 2008)]." However Pengornis is an early Cretaceous taxon--please revise this section, and note that Chiappe (among others) highlighted the fact that Avisaurids are relatively large birds. This is stated on several papers, including those cited in this papers.

3. The discussion about a trend of increased body size during the evolutionary history of enantiornithines needs a bit more of historical context, as the authors--based on the references they cite--make it look as this has only been noticed recently. In fact, almost 30 years ago (e.g. Chiappe 1996) this trend was clearly noticed together with the fact that the entire spectrum of body size increased significantly towards the Late Cretaceous. As an example, I'm including a section from Chiappe & Walker (2002), a paper already cited by the authors:

"During their evolution throughout the Cretaceous there is a general increment in size and size range among the eu- enantiornithine species. All known Early Cretaceous eu- enantiornithines are small to medium-size birds (see com- parative measurements in Bochenski, 1996). Sinornis and Eoalulavis, for example, were the size of a sparrow or ﬁnch, and the size of Nanantius and Concornis was comparable to that of a medium-sized thrush. Late Cretaceous euenan- tiornithines, however, are generally larger: Gobipteryx was the size of a jackdaw and Neuquenornis that of a small fal- con. Some of the Late Cretaceous taxa were much larger. Enantiornis had a wingspan of about 1.2 meters, ranging in size between a skua and a turkey vulture (Chiappe, 1996b), and a recently discovered enantiornithine from France has been compared in size to a herring gull (Buffetaut, 1998). Some Late Cretaceous euenantiornithines, however, re- tained the small size of their Early Cretaceous relatives; Alexornis was sparrow-sized."

I'm happy to provide the specific citations but I'm sure the authors can find them easily and cite them.

Luis Chiappe

6. PLOS authors have the option to publish the peer review history of their article (what does this mean?). If published, this will include your full peer review and any attached files.

Reviewer #1: No

Reviewer #2: **Yes: **Luis M. Chiappe

---

## [Author Response · Author response to Decision Letter 0]

9 Aug 2024

Editor

 Completed

2. In your manuscript, please provide additional information regarding the specimens used in your study. Ensure that you have reported human remain specimen numbers and complete repository information, including museum name and geographic location. 

For more information on PLOS ONE's requirements for paleontology and archeology research, see https://journals.plos.org/plosone/s/submission-guidelines#loc-paleontology-and-archaeology-research.

Completed

3. Please take this opportunity to be sure you have met all of our guidelines for new species. For proper registration of a new zoological taxon, we require two specific statements to be included in your manuscript.

1) In the Results section, the globally unique identifier (GUID), currently in the form of a Life Science Identifier (LSID), should be listed under the new species name, for example:

Anochetus boltoni Fisher sp. nov. urn:lsid:zoobank.org:act:B6C072CF-1CA6-40C7-8396-534E91EF7FBB

Another LSID for the manuscript itself should also appear within the Nomenclature statement. You will need to contact Zoobank (zoobank.org/About) to obtain a GUID (LSID). You should receive one LSID for your manuscript and a separate, unique LSID for the new species. 

 Completed

2) Please also insert the following text into the Methods section, in a sub-section to be called ""Nomenclatural Acts"":

 Completed

The electronic edition of this article conforms to the requirements of the amended International Code of Zoological Nomenclature, and hence the new names contained herein are available under that Code from the electronic edition of this article. This published work and the nomenclatural acts it contains have been registered in ZooBank, the online registration system for the ICZN. The ZooBank LSIDs (Life Science Identifiers) can be resolved and the associated information viewed through any standard web browser by appending the LSID to the prefix ""http://zoobank.org/"". 

The LSID for this publication is: urn:lsid:zoobank.org:pub: XXXXXXX. 

The electronic edition of this work was published in a journal with an ISSN, and has been archived and is available from the following digital repositories: PubMed Central, LOCKSS [author to insert any additional repositories].

 Completed

All PLOS ONE articles are deposited in PubMed Central and LOCKSS. If your institute, or those of your co-authors, has its own repository, we recommend that you also deposit the published online article there and include the name in your article.

Following a recent ruling by the International Commission on Zoological Nomenclature, electronic journals are now a valid format for publication of new zoological taxa. In order to ensure the valid publication of your new species, please be sure to include the updated version of Nomenclatural Acts (above). 

A complete explanation of our guidelines for publishing new species can be found on our website: http://www.plosone.org/static/guidelines#zoological.

 Completed

5. Please amend either the title on the online submission form (via Edit Submission) or the title in the manuscript so that they are identical.

 Completed

6. We are unable to open your Supporting Information file [AvisaurusTMT_Rcode (for journal).R]. Please kindly revise as necessary and re-upload.

 Completed

 Completed

Reviewer #1: Clark et al. describe three new specimens of avisaurids birds, and use them to support the existing hypothesis that avisaurids were raptorial birds. Attached to my review is a PDF of the manuscript with comments made directly.

The comments made throughout the manuscript by the reviewer were both helpful and productive, and we thank the reviewer for their advice and assistance in creating a stronger manuscript. We found the section concerning the mass ranges of birds particularly helpful and constructive. We have made sure that our wording is clearer when referring to a particular group or when referring to the higher percentile range of that particular group [i.e., we have made it clear when referring to a portion of a family (“larger members”) or to the family as a whole]. We have also included better and more specific sources for where size ranges are sourced from. 

All comments or suggestions made by the reviewer which required a simple fix or alteration have been corrected throughout the manuscript and figures. There were a number of points brought up within the manuscript notes that we will address here. Lastly, we have also revised all figures as requested including figure 6, 8, 11 and included a new supplemental figure with better images of the proximoplantar fossa as requested (Fig. S2). 

Comparison with penguins: We removed the brief section in the discussion that covered comparisons with penguins as requested. Originally, we wanted to include a brief section comparing this group as we felt that including them made sense given the superficially similar aspect ratios. However, as commented on, the manuscript does not require this section and it has since been removed to make the discussion more streamline. 

Pengornathids as late-diverging enantiornthines: Thank you for pointing this phylogenetic result out! Our results do differ from those in Atterholt et al., (2018), but pengornithids are recovered as late-diverging enantiornithines in other published literature as well:

O’Connor et al., 2013 - 10.1080/02724634.2012.719176

Wang et al., 2022 - 10.1017/jpa.2022.12 

Li et al., 2021 - 10.1111/joa.13588 

O’Conner et al., 2020 - 10.1038/s41586-020-2945-x 

Introduction comment: We find that the section within the introduction which discusses previous criteria for the Avisauridae is needed both for the reader to understand the history of avisaurids, and for the introduction to be a proper review of previous literature. For readers with limited knowledge of this clade of enantiornithines, we find that introducing these topics (discussing prior synapomorphies) within the introduction to be important and relevant. By discussing these taxa within the introduction, we feel that we have given the reader a thorough and relevant summary of the current state of this clade of enantiornithines. Additionally, we briefly cover why there are initial issues with previous criteria for admittance to this clade. With this section in the introduction, it allows the reader to see that prior criteria for consideration into the Avisauridae requires an update – which is something we state as an objective in the introduction, and also why some taxa are not considered to be avisaurids here. 

Figure 6: With regards to additional work for figure 6 (regression analysis): We have made sure to include both slope and resulting R2 values for both A and B in figure 6 (and for figure S1 were requested). However, we find that phylogenetically influenced results when dealing with enantiornithines can lead to further confusion given that the phylogeny of this entire clade is still highly debated. Our objective in this figure is to only direct the reader to the fact that over time, as far as we can tell from the fossil record, enantiornithine body size remains fairly conserved early in their evolution and expands to both lower and higher thresholds by the end Cretaceous. 

Line 545 (relating to lowering aspect ratio equating to greater efficiency) “This is untrue. Chiefly, a lower aspect ratio does not inherently affect the relative length of the tmt. Aspect ratio chiefly describes the relative length and width within the tmt, but not its muscle tendon insertion location nor its length relative to any other bone in the hindlimb. If an accipitrid had its tmt midshaft stretched laterally its aspect ratio would reduce, but the tmt would still be just as long and have the same fulcrum, moment of action, and resistance locations.”

 In birds, we typically do see that lower aspect ratio tarsometatarsi do change the proportional lengths against the tibiotarsus. Additionally, when the aspect ratio diminishes, particularly in birds of prey, we also see an increase in the proportional size (i.e., surface area) of the tubercle. In the hypothetical example given by the reviewer (stretching a hawk’s tarsometatarsus), we would expect, based on all other birds of prey, that the relative position and surface area of the tubercle would also change, creating a whole different lever system. If the tarsometatarsus was stretched mediolaterally, it would alter how the hindlimb works and in reality, other parts of the system would change. To the point, we see this very thing happen but in reverse in the Barn owl (Tyto alba). Barn owls have essentially taken the owl-like tarsometatarsus and increased the aspect ratio, elongating it. The result is not little change to the lever system, but alternatively, it has taken the form of a much more hawk-like tarsometatarsus, resulting in an altogether different lever system than other strigiforms. In this relevant example, by changing the aspect ratio, you change the relative length compared to the tibiotarsus, and the position of the tubercle. These results are actually forthcoming in another manuscript soon to be submitted focused on accipitrid and strigid tarsometatarsi. 

 We find that changes in aspect ratios do in fact change relative location of the tubercle and the lever system as a whole. Lastly, by lowering the aspect ratio, you inherently reduce the proportional length against the other hindlimb bones among birds. Aspect ratio does not scale proportionally among birds of prey – which is made evident in owls and hawks assuming different methods of prey acquisition. Again, when a member of either order evolves aspect ratios closer to the other order, it then exhibits prey acquisition strategies closer to the latter. 

Comment: “Perhaps the most obvious association with a higher aspect ratio goes unmentioned, though: Higher resistance to bending forces. As elongate bones thicken in a dimension perpendicular to their long axis they tends to better resist bending forces (see e.g. Cuff and Rayfield). So a tmt with a relatively low aspect ratio will be relatively better able to resist forces like those in locomotion and hunting”.

 Thank you for bringing this point up. This thought did occur to us during our initial research, however, a key feature that differs between neornithines and enantiornithines kept this point from our discussion. In neornithines rigidity of the foot is far better facilitated as they have fully fused metatarsals while enantiornithines do not. Even robust avisaurids still retain the enantiornithine feature of fusion limited to the proximal portion. Torsion of the individual metatarsals would be more likely for enantiornithines compared to extant birds of prey today. However, as the reviewer states, this point is debatable. However, we feel it would not contribute enough to be included in our discussion due to the high level of uncertainty regarding rigidity in unfused metatarsals. 

Owl vs hawk comments: There appeared to a few points of misunderstanding with the reviewer that we believe were both addressed in the original draft and are further clarified in the revised manuscript. Particularly with the comments addressing lines 539 and 545 (above), we felt that we made it clear that avisaurids did not function either as owls or as accipitrids in their given environment, but they do display morphological similarities to both groups, suggesting a unique morphological solution to perhaps a similar predatory niche. We lean more so into strigids throughout our discussion because they show more similarities with the lower aspect ratio of the tarsometatarsus and the subsequent lever system implications. However, we have made sure throughout the revised manuscript to state that we are drawing from strigid morphology, not stating it as a direct analogue for how avisaurids functioned. We have more our discussion points, including the ability to lift proportionally heavy things, clearer and address limitations of increasing size. Thank you for making this clearer. 

Line 553. “These are not just driven by the low aspect ratio of the tmt, though. The cranial face of owl tarsometatarsi have deep excavations that run down nearly half the length for additional adductor muscle attachment area, and the plantar side is even more dramatically incised to accommodate the large flexor muscles. While the low aspect ratio of the avisaurid tmt does allow for relatively broader muscle attachments than other enantiornithines, it seems unlikely to me that gripping abilities analagous to owls can be attributed to Avisauridae.”

 We agree with the reviewer; however, we do not state that avisaurids had the gripping abilities as owls – only that they were likely increased relative to other enantiornithines (which we have clarified in the text). We have made sure to include words such as “analogous” to strigid grip where requested in the manuscript. The objective of this portion of the discussion was to signal that owls have morphologies that allow for increased grip relative to accipitrids. In the same way, avisaurids show similar, uniquely-strigid reminiscent morphologies. 

Fig. 9 comment: “This is likely more an effect of body size than musculature. Lift scales with wing area and muscle cross-sectional area, both of which scale to body mass^2/3. As such, larger raptorial birds tend to either hunt relatively less massive prey or to kill prey on the ground without lifting it. The Northern pygmy owl in the figure has an average mass of around 70 g, less than a tenth of mass estimates for A. darwini. So feats like this would be unlikely for the avisaurid.”

 We agree with the reviewer. Assuredly, a bird the size of a large avisaurid could not lift items nearly three times its own weight (similar to much smaller strigids). However, the biomechanics for increased compensation for carrying weight are present. We have made sure to make this clear in the figure’s description and the discussion/conclusion

---

## [Editor Report · Decision Letter 1]

26 Aug 2024

PONE-D-24-29019R1New enantiornithine diversity in the Hell Creek Formation and the functional morphology of the avisaurid tarsometatarsusPLOS ONE

Dear Dr. Clark,

Thank you for submitting your manuscript to PLOS ONE. After careful consideration, we feel that it has merit but does not fully meet PLOS ONE’s publication criteria as it currently stands. Therefore, we invite you to submit a revised version of the manuscript that addresses the points raised during the review process.

We look forward to receiving your revised manuscript.

Kind regards,

Felipe Lima Pinheiro, Ph.D

Academic Editor

PLOS ONE

Journal Requirements:

Additional Editor Comments:

Dear Alexander,

Thank you very much for the revised version of your manuscript. You have successfully addressed all the issues raised by the reviewers.

Before final acceptance, I have just a few editorial comments, which you can find in the attached PDF. I believe these last modifications will not require much time or effort.

---

## [Author Response · Author response to Decision Letter 1]

27 Aug 2024

Dear Editor,

Thank you again for your assistance and throughout read-through of our manuscript. I have addressed your final comments here. Thank you again very much!

Sincerely, 

Alex Clark 

Changes made:

Thomas Carr was removed as an author – I have received confirmation from all authors that they responded to the journal in agreement of the change.

A reference has been added that was previously cited but missing the citation:

Schneider, C.A., Rasband, W.S., Eliceiri, K.W., 2012. NIH Image to ImageJ: 25 years of image analysis. Nat Methods 9, 671–675. https://doi.org/10.1038/nmeth.2089

Editor: 

Please indicate which results are derived exclusively from the majority-rule tree. Personally, I have some reservations about accepting taxonomic decisions and reevaluations that are not well supported. It seems to me that some redefinitions are being based exclusively on the majority-rule tree, correct? Do the authors wish to maintain this approach? If so, I suggest indicating potential weaknesses inherent to this practice.

Thank you for pointing this out. We have since clarified this results section concerning characters, source of synapomorphies, and where similar morphological characters support inclusion within the clade. 

I do believe that the correct plural term is Lagerstätten. Please change throughout.

 According to a previous reviewer and internet sources, Lagerstätte is the plural. When entered into search fields as a plural noun, it is also corrected as Lagerstätte.

---

## [Editor Report · Decision Letter 2]

4 Sep 2024

New enantiornithine diversity in the Hell Creek Formation and the functional morphology of the avisaurid tarsometatarsus

PONE-D-24-29019R2

Dear Dr. Clark,

We’re pleased to inform you that your manuscript has been judged scientifically suitable for publication and will be formally accepted for publication once it meets all outstanding technical requirements.

Kind regards,

Felipe Lima Pinheiro, Ph.D

Academic Editor

PLOS ONE
---

## [Editor Report · Acceptance letter]

16 Sep 2024

PONE-D-24-29019R2 

PLOS ONE

Dear Dr. Clark, 

I'm pleased to inform you that your manuscript has been deemed suitable for publication in PLOS ONE. Congratulations! Your manuscript is now being handed over to our production team.

Kind regards, 

on behalf of

Dr. Felipe Lima Pinheiro 

Academic Editor

PLOS ONE